# XIMP: CROSS GRAPH INTER-MESSAGE PASSING FOR MOLECULAR PROPERTY PREDICTION

## ABSTRACT

Accurate molecular property prediction is central to drug discovery, yet Graph Neural Networks (GNNs) often underperform in data-scarce regimes and can trail fixed fingerprints. We introduce XIMP (Cross Graph Inter-Message Passing), which performs message passing both *within* and *across* graph representations, integrating multiple granularities in a single model. In our chemistry setting, XIMP unifies the molecular graph with junction trees (scaffold-aware) and extended reduced graphs (pharmacophoric), enabling per-atom use of complementary views and exceeding the expressivity of the Weisfeiler-Leman isomorphism test. Unlike prior work – often limited to few abstractions, indirect exchange via the original graph, or overlooking oversquashing – XIMP supports arbitrary numbers of abstractions and both direct as well as indirect inter-abstraction communication. Across ten diverse molecular property-prediction tasks, XIMP outperforms state-of-the-art GNNs and fingerprint baselines in most cases, leveraging interpretable abstractions as an inductive bias to guide learning toward established chemical concepts and enhance generalization in data-scarce regimes.

## 1 INTRODUCTION

Early drug discovery benefits from machine learning; classical models (e.g., SVMs, random forests) have long been used for molecular data (Dara et al., 2022). Recently, deep learning has driven advances in image recognition (Krizhevsky et al., 2017), natural language processing (OpenAI, 2024), and the life sciences (Jumper et al., 2021), enabled by large datasets and efficient processing. A core task is molecular property prediction, where compounds are represented by chemical fingerprints encoding substructures as bit vectors. Variants that extract and encode molecular subgraphs exist (Rogers & Hahn, 2010; Daylight, 2008), but their predictive performance varies across tasks.

Graph neural networks (GNNs) are widely applied to molecular property prediction (Chami et al., 2022; Gilmer et al., 2017), but on key tasks they often fail to surpass classical fingerprint-based methods (Stepišnik et al., 2021; Jiang et al., 2021), with gains mainly on large datasets (Jiang et al., 2021; Deng et al., 2023). Many tasks lack such data. A common remedy is injecting domain knowledge via chemical structure augmentation (Magar et al., 2022) or invariant/equivariant architectures (Cremer et al., 2023). These *inductive biases* align models with chemical principles, boosting generalization in low-data regimes: equivariant models enforce physical symmetries, while augmentations expose identity-preserving variations. Reduced graphs provide a complementary bias by abstracting molecules into functional groups, rings, and pharmacophores, emphasizing features central to recognition and activity. Embedding such abstractions into neural architectures promotes interpretability, reduces data demands, and strengthens robustness and predictive performance. However, the integration of interpretable reduced graphs and their cross-communication within GNNs remains underexplored. In chemistry, multiple abstractions are often (i) processed sequentially (Kong et al., 2022), or, when simultaneous, either (ii) message passing is limited to local aggregation (Ji et al., 2022), (iii) communication occurs only indirectly via the molecular graph (Li et al., 2024), or (iv) only a single abstraction is used (Fey et al., 2020; Li et al., 2024; Kong et al., 2022). Outside chemistry, Finder et al. (2025) show that direct cross-graph communication mitigates oversquashing and improves long-range modeling, albeit with generic abstractions. Moreover, Deep graph matching (Wang et al., 2018; Li et al., 2019), as well as hierarchical pooling (Ying et al., 2018) also exchange information across graphs, but address tasks other than molecular property prediction.

To address this gap, we propose XIMP (cross graph inter-message passing), a versatile architecture that integrates multiple graph views at different coarseness levels via intra- and inter-graph message passing. We combine the molecular graph with two complementary abstractions: Junction Trees (JT) (Rarey & Dixon, 1998), capturing hierarchical fragment organization, and extended reduced graphs (ErG) (Stiefl et al., 2006), encoding pharmacophoric features and topology. We further coarsen JTs to mitigate oversquashing and better capture long-range interactions. By explicitly linking these views, XIMP learns chemically informative representations where conventional GNNs fail.

Empirically, we benchmark XIMP against various GNN baselines and its closest competitor, which either (i) fuses multiple representations only at the final layer or (ii) performs inter-message passing for a single graph pair (Fey et al., 2020). XIMP typically outperforms these state-of-the-art models across diverse molecular property prediction tasks, effectively leveraging complementary graph abstractions. XIMP also surpasses fixed descriptors such as ECFP (Rogers & Hahn, 2010), underscoring the benefit of learning from chemically meaningful abstractions.

**Related Work.** In principle, most GNNs, e.g., (Kipf & Welling, 2017; Li & Leskovec, 2022; Hamilton et al., 2017), can operate on molecular graphs, but conventional message passing is generally limited by the Weisfeiler-Leman (1-WL) test and not universal for graph functions (Xu et al., 2019; Weisfeiler & Leman, 1968). They can even fail on tasks such as small-cycle detection (Chen et al., 2020), crucial for properties tied to local structures (e.g., aromatic rings). Remedies include invariant graph networks (Maron et al., 2019a;b), relational pooling (Murphy et al., 2019; Chen et al., 2020), and higher-order WL extensions (Morris et al., 2019). In contrast, hierarchical inter-message passing (HIMP) is a simple approach with state-of-the-art performance across multiple datasets (Fey et al., 2020). Applied to JTs, it distinguishes molecules such as decalin and bicyclopentyl – indistinguishable by 1-WL on the molecular graph – by exploiting their 1-WL-distinguishable reduced forms, thereby capturing subtle, biologically relevant variations.

However, integrating chemically interpretable reduced graphs and modeling their communication within message passing remains underexplored. JTs have primarily supported molecule generation (Jin et al., 2019; 2018). For property prediction, RG-MPNN (Kong et al., 2022) applies an ErG-like reduction but processes graphs sequentially, unlike HIMP's simultaneous scheme. Recent works employ multiple reductions (JT, ErG, functional-group graphs (Ji et al., 2022)) but restrict message passing to each graph and pool via a super-node (Kengkanna & Ohue, 2023; 2024). Neural atoms (Li et al., 2024) allow molecular-reduced graph messaging, but the size of the reduced graph is manually fixed. Generic reductions show that inter-graph passing mitigates oversquashing and improves long-range performance (Finder et al., 2025). Similarly, Wollschläger et al. (2024) introduce substructure-aware biases, though limited to a single fragment abstraction and neighbor-only communication. Their bias is mainly topological (rings, paths, junctions), lacking explicit aromaticity, pharmacophoric rules, or electronic effects. Thus, while multiple chemical abstractions have been used, prior work does not fully exploit inter-graph communication.

Conversely, prior work on inter-graph information exchange addresses different problems. Deep graph matching (Wang et al., 2018; Li et al., 2019), compares structural similarity between distinct graphs via cross-graph message passing, unlike our setting, which exchanges messages among representations of the *same* graph. Graph pooling methods (e.g., DIFFPOOL (Ying et al., 2018), $top_k$ pooling (Gao & Ji, 2022)) learn coarser representations, where inter-message passing aggregates information so that representative nodes capture key signals for downstream layers.

**Contributions.** We introduce two variants of inter-message passing that enable information exchange between a graph and an arbitrary number of its abstractions. Unlike existing techniques, (i) communication can occur simultaneously between the graph and *any* number of abstractions *and* among the abstractions themselves, and (ii) both *direct* and *indirect* communication pathways are considered. While domain-agnostic and not technically limited to hierarchical representations or chemistry, we demonstrate XIMP's benefits on molecular tasks, where it learns from chemically and structurally interpretable abstractions, for which we also characterize its enhanced expressivity beyond the 1-WL test. Moreover, we show XIMP's potential to mitigate oversquashing and improve long-range interactions by using multiple *coarseness* levels in the structural abstractions. Together, these components yield XIMP, a generalization of prior inter-message passing methods that performs strongly compared to state-of-the-art baselines accross various datasets and tasks.

## 2 PRELIMINARIES

In this section, we discuss graph abstractions for molecules, neural architectures using single and multiple graphs, and introduce our notation along the way.

**Molecular Graphs and Reduced Graphs.** A molecular graph encodes a molecule's structure: nodes represent the atoms and edges the bonds (Figure 1a (b)). As XIMP can operate on multiple graph views concurrently, this includes *reduced graphs* (Birchall & Gillet, 2011) extracted from the molecular graph. A reduced graph modifies the structure while retaining and/or adding information about key features and topology, enabling more generalized analyses of the represented molecule. We briefly describe two complementary reduced graphs that provide interpretable abstractions matching our chemistry setting. (i) The *junction tree* (JT) (Jin et al., 2018), also used in HIMP, is a hierarchical tree of molecular fragments whose nodes represent bonds, rings, bridges, or singletons (Figure 1a (d)). (ii) The *extended reduced graph* (ErG) (Stiefl et al., 2006) encodes pharmacophoric features and atom-level topological relationships relevant to biological activity (Figure 1b). JTs and ErGs provide abstract molecular representations at different granularities. JTs condense the molecular graph into a minimal set of fragment nodes, enabling long-range interaction modeling. ErGs, on the other hand, are more fine-grained: they treat rings with explicit bridgehead handling and add node features encoding higher-level pharmacophoric information absent in JTs and standard molecular graphs. Together with the molecular graph, these views offer complementary, non-redundant perspectives on molecular topology. A detailed description is available in Appendix B.

**Message Passing Graph Neural Networks.** GNNs learn node representations through *message passing* (MP), which propagates information across nodes and edges (Gilmer et al., 2017). They show strong performance across domains, including social networks (Wu et al., 2020), drug discovery (Chami et al., 2022), and others (Dai et al., 2017). In molecular property prediction, GNNs operate on a molecular graph $G = (V, E)$, where $V$ is a set of atoms and $E$ is a set of bonds of a molecule. First, each node $v \in V(G)$, and edge $(u, v) \in E(G)$ is endowed with an initial feature vector $\boldsymbol{x}_u^{(0)} \in \mathbb{R}^{d_v}$ and $\boldsymbol{e}_{(u,v)}^{(0)} \in \mathbb{R}^{d_e}$ representing, for example, the atom and bond type, respectively. The node embeddings are then updated through message-passing layers by aggregating information from their neighbors $N(v)$. The $l$-th layer recursively updates the embedding $\boldsymbol{x}_v^{(l)}$ of node $v$ via

$$\boldsymbol{m}_v^{(l)} = \text{AGG}_{\boldsymbol{\theta}_1^{(l)}}\Big(\big\{\!\!\big\{ \big(\boldsymbol{x}_w^{(l-1)}, \boldsymbol{e}_{(w,v)}^{(l-1)}\big) \;\big|\; w \in N(v) \big\}\!\!\big\}\Big), \quad \boldsymbol{x}_v^{(l)} = \text{COMB}_{\boldsymbol{\theta}_2^{(l)}}\Big(\boldsymbol{x}_v^{(l-1)}, \boldsymbol{m}_v^{(l)}\Big),$$

where $\boldsymbol{e}_{(w,v)}^{(l)}$ represents the embedding of edge $(w, v) \in E(G)$, $\{\!\!\{ \cdot \}\!\!\}$ denotes a multiset and $N(v) = \{u \in V(G) \mid (u, v) \in E(G)\}$ are the neighbors of the node $v$. The functions AGG and COMB are parameterized by $\boldsymbol{\theta}_1^{(l)}$ and $\boldsymbol{\theta}_2^{(l)}$, respectively, which are optimized during training. Ultimately, after passing through $L$ layers, the output of the READ function provides the graph embedding optimized for predicting the desired molecular property:

$$\boldsymbol{h}_G = \text{READ}\Big(\big\{\!\!\big\{ \boldsymbol{x}_v^{(L)} \mid v \in V(G) \big\}\!\!\big\}\Big).$$

With multiple graph abstractions, *intra-message passing* restricts communication to a single graph; message passing runs in parallel on the molecular graph and on each abstraction. In contrast, *inter-message passing* enables cross-graph exchange: information flows between the molecular graph and its abstractions and, in XIMP, also between different abstractions.

**Hierarchical Inter-Message Passing.** HIMP (Fey et al., 2020) is a GNN architecture that leverages two GNN models. One model operates on the molecular graph (GIN-E), while the other operates on its corresponding JT (GIN), leveraging the hierarchical representation. Both models use message passing as described in Section 2. Additionally, after each layer, an *inter-message passing (IMP)* model facilitates the exchange of information between these two graph representations (Figure 2). For a complete outline of HIMP, see Appendix G. HIMP and XIMP instantiate message passing with GIN-E (Graph Isomorphism Network with edge features) on molecular graphs with edge features (Hu et al., 2020), and with GIN on abstractions without edge features (e.g., JTs, higher-order structures (Li & Leskovec, 2022)); details in Appendix F.

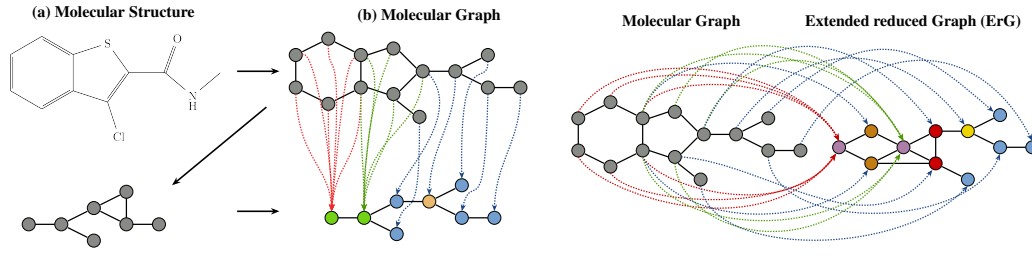

**(a)** The junction tree decomposition converts a molecular structure (a) into a molecular graph (b), where nodes are atoms and edges are bonds. A cluster graph (c) is built by grouping atoms into cycles and assigning clusters to non-cycle edges, with edges between clusters if they share atoms. Cycles in the cluster graph are removed by adding the shared atom as a separate cluster, yielding the junction tree (d).

**(b)** Molecular graph (left) and extended reduced graph (right). Construction: (1) adjust atom charges to reflect physiological conditions; (2) assign H-bond donor/acceptor properties; (3) identify and tag endcap groups (lateral hydrophobic features, including thioethers); (4) add a ring centroid (*aromatic/hydrophobic*), link it to substituted atoms/bridgeheads, remove unsubstituted atoms, and keep bonds among the rest.

Figure 1: Molecular graph abstractions: (a) junction tree and (b) extended reduced graph. Arrows indicate node mappings between the graphs; their colors encode singleton or group memberships.

## 3 CROSS GRAPH INTER-MESSAGE PASSING

We generalize existing inter-message passing approaches to an arbitrary number of abstractions and direct as well as indirect communication flows. To that purpose, we introduce two variants of inter-message passing incorporating reduced graphs.

**Indirect Inter-Message Passing ($I^2$MP).** Information exchange between the original (molecular) and other graph representations occurs within each layer. While we call these other representations *reduced*, they can be arbitrary graph representations.

Formally, let $\boldsymbol{X}^{(l)} \in \mathbb{R}^{|V(G)| \times d_x}$ be the embedding matrix of the molecular graph $G$ in layer $l$, and $\boldsymbol{T}_1^{(l)}, \boldsymbol{T}_2^{(l)}, ..., \boldsymbol{T}_n^{(l)}$ be the embedding matrices of the corresponding other representations (reduced graphs) $T_1, T_2, ..., T_n$ in layer $l$. Note that, $\boldsymbol{T}_i^{(l)} \in \mathbb{R}^{|V(T_i)| \times d_x}$ for all $i$ in $\{1, 2, \ldots, n\}$. The inter-message passing step changes the intermediate embedding matrices as follows:

$$\widetilde{\boldsymbol{X}}^{(l)} \leftarrow \boldsymbol{X}^{(l)} + \sum_{i=1}^{n} \sigma\left(\boldsymbol{S}_i \boldsymbol{T}_i^{(l)} \boldsymbol{W}_{i,1}^{(l)}\right), \quad \widetilde{\boldsymbol{T}}_i^{(l)} \leftarrow \boldsymbol{T}_i^{(l)} + \sigma\left(\boldsymbol{S}_i^T \boldsymbol{X}^{(l)} \boldsymbol{W}_{i,2}^{(l)}\right),$$

where $\boldsymbol{W}_{i,1}^{(l)}, \boldsymbol{W}_{i,2}^{(l)} \in \mathbb{R}^{d_x \times d_x}$ are trainable parameters within layer $l$, $\boldsymbol{S}_i \in \{0,1\}^{|V(G)| \times |V(T_i)|}$ are the mapping matrices that store the assignment of nodes of the molecular graph to nodes of the $i$-th reduced graph, and $\sigma$ denotes non-linearity. It is important to note that for our purposes, $\boldsymbol{X}$ must be the embedding matrix of $G$ since we require a mapping between $\boldsymbol{X}$ and all $\boldsymbol{T}_i$.

**Direct Inter-Message Passing (DIMP).** Inter-message passing can also be done directly between reduced graphs. Beyond exchanging information between the original molecular graph $G$ and each reduced graph $T_i$, we also allow *pairwise* communication among the reduced graphs. Let $\mathbf{T}_i^{(l)} \in \mathbb{R}^{|V(T_i)| \times d_x}$ be the layer-$l$ embeddings of $T_i$. We assume that every node of $G$ belongs to *at least one* node in each reduced graph $T_i$ (i.e., there exists a left-total abstraction assignment $R_i \subseteq V(G) \times V(T_i)$), while one node of $T_i$ may summarize many nodes of $G$. Hence, in the assignment matrix $\boldsymbol{S}_i \in \{0,1\}^{|V(G)| \times |V(T_i)|}$ each row contains one or more 1s (or, more generally, weighted memberships). This formulation naturally accounts for overlapping abstractions such as fused rings or functional groups, where a node of $G$ can contribute to multiple reduced nodes. For every unordered pair $(i, k)$ with $i \neq k$, messages from $T_k$ to $T_i$ are computed by

$$\mathbf{M}_{k \to i}^{(l)} = \sigma\left(\widetilde{\boldsymbol{S}}_{ik} \mathbf{T}_k^{(l)} \boldsymbol{W}_{k \to i}^{(l)}\right), \qquad \widetilde{\boldsymbol{S}}_{ik} = \boldsymbol{D}_{T,i}^{-1} \boldsymbol{S}_i^\top \boldsymbol{D}_{G,k}^{-1} \boldsymbol{S}_k,$$

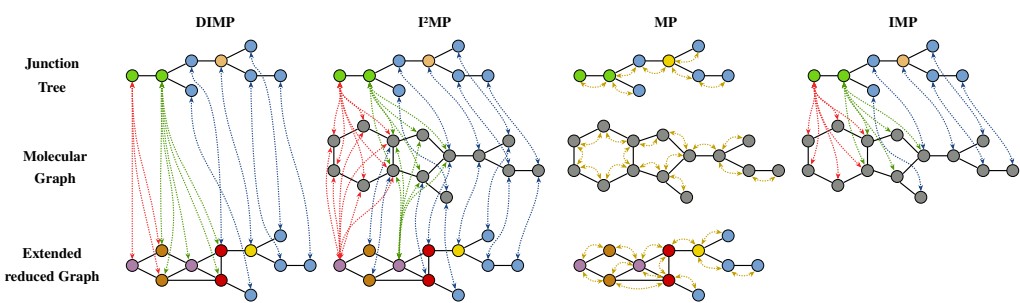

Figure 2: Visualization of communication flows in XIMP and HIMP. XIMP employs DIMP, I²MP, and MP, while HIMP uses only IMP and MP. Node colors denote graph semantics; arrows indicate bidirectional message passing: red/green for one-to-many ring–abstraction relations, blue for one-to-one cross-graph relations, and yellow for one-to-one within-graph relations.

where $\boldsymbol{W}_{k\to i}^{(l)} \in \mathbb{R}^{d_x \times d_x}$ are trainable, $\sigma$ is a non-linearity (ReLU), $\boldsymbol{D}_{T,i} = \text{diag}(\boldsymbol{S}_i^\top \mathbf{1})$ stores the sizes of the $T_i$ clusters (total membership of nodes of $G$ in each reduced node), and $\boldsymbol{D}_{G,k} = \text{diag}(\boldsymbol{S}_k \mathbf{1})$ stores the number of nodes of $T_k$ assigned to each node of $G$. The left normalizing matrix $\boldsymbol{D}_{G,k}^{-1}$ ensures that a node of $G$ with multiple memberships in $T_k$ does not contribute disproportionately when projecting from $T_k$ to $G$, while the right normalizing matrix $\boldsymbol{D}_{T,i}^{-1}$ balances the aggregation of multiple nodes of $G$ into a node of $T_i$. This prevents multiplicity-induced overweighting, ensures stable message magnitudes across overlaps, and reflects the chemical intuition that atoms participating in several functional roles (e.g., ring junctions or substituents) should distribute their influence fairly among all abstractions. We formalize this as follows (proof in Appendix J):

**Proposition 3.1.** *Let $T_k, T_i$ be left-total abstractions of $G$, with $\boldsymbol{S}_k \in \{0,1\}^{|V(G)| \times |V(T_k)|}$, $\boldsymbol{S}_i \in \{0,1\}^{|V(G)| \times |V(T_i)|}$ and $\widetilde{\boldsymbol{S}}_i = \boldsymbol{D}_{T,i}^{-1} \boldsymbol{S}_i^\top$ and $\widetilde{\boldsymbol{S}}_k = \boldsymbol{D}_{G,k}^{-1} \boldsymbol{S}_k$ such that $\widetilde{\boldsymbol{S}}_{ik} = \widetilde{\boldsymbol{S}}_i \widetilde{\boldsymbol{S}}_k$. Further, let $\widetilde{\mathbf{M}}_{k\to i}^{(l)} = \widetilde{\boldsymbol{S}}_{ik} \mathbf{T}_k^{(l)}$ be the messages from $T_k$ to $T_i$ before the application of the trainable parameter matrix or a nonlinearity. Then, the following statements hold:*

*1. $\|\widetilde{\mathbf{M}}_{k\to i}^{(l)}\|_\infty \leq \|\mathbf{T}_k\|_\infty$ and $\|\widetilde{\mathbf{M}}_{k\to i}^{(l)}\|_1 \leq \alpha \|\mathbf{T}_k\|_1$ for some $\alpha \in \mathbb{R}$, $\frac{|V(T_i)|}{|V(T_k)|} \leq \alpha \leq |V(T_i)|$*

*2. For any $x \in \mathbb{R}^{d_x}$ and $\boldsymbol{T}_k = \mathbf{1} \, x^\top \in \mathbb{R}^{|V(T_k)| \times d_x}$, $\widetilde{\boldsymbol{S}}_{ik} \, \boldsymbol{T}_k = \mathbf{1} \, x^\top$*

Next, reduced-graph embeddings are updated additively:

$$\widetilde{\mathbf{T}}_i^{(l)} \leftarrow \mathbf{T}_i^{(l)} + \mathbf{M}_{k\to i}^{(l)}, \qquad \widetilde{\mathbf{T}}_k^{(l)} \leftarrow \mathbf{T}_k^{(l)} + \mathbf{M}_{i\to k}^{(l)},$$

for all $i, k \in [1, n]$ with $i \neq k$. Note that if $T_k$ is a partition of $G$, then $\boldsymbol{D}_{G,k} = \boldsymbol{I}$ and the expression reduces to the simpler form $\widetilde{\boldsymbol{S}}_{ik} = \boldsymbol{D}_{T,i}^{-1} \boldsymbol{S}_i^\top \boldsymbol{S}_k$. Finally, we define our READ function as the combination of mean READ of individual graph representations, each weighted by a trainable matrix:

$$h_G = \frac{1}{|V(G)|} \sum_{i=1}^{|V(G)|} \boldsymbol{x}_i^{(L)} \boldsymbol{W}_0 \bigoplus_{j=1}^{n} \frac{1}{|V(T_j)|} \sum_{i=1}^{|V(T_j)|} \boldsymbol{t}_{j,i}^{(L)} \boldsymbol{W}_j$$

with $\bigoplus$ denoting a graph level aggregation (i.e., concatenation $||$ or summation $\sum$), $\boldsymbol{W}_0, \boldsymbol{W}_1, ..., \boldsymbol{W}_n \in \mathbb{R}^{d_x \times d_x}$ are trainable matrices, $\boldsymbol{x}_i^{(L)} \in \mathbb{R}^{d_x}$ holds for the final embedding of the $i$-th node of graph $G$ after $L$ layers, similarly $\boldsymbol{t}_{j,i}^{(L)} \in \mathbb{R}^{d_x}$ denotes the final embedding of the $i$-th node of the $j$-th reduced graph.

**Mitigating Oversquashing.** Oversquashing was first linked to bottleneck edges (Alon & Yahav, 2021) and later to inter-node distances (Di Giovanni et al., 2023). A complementary perspective views oversquashing as a manifestation of limited effective receptive field in GNNs (Finder et al., 2025). Many molecular properties (e.g., solubility, lipophilicity) depend on long-range interactions, not just local groups; oversquashing hinders this propagation. Using XIMP, we mitigate this with an approach motivated by, yet orthogonal to, Finder et al. (2025): instead of changing the propagation scheme, we coarsen the JT by repeatedly contracting leaves into parents, shortening paths and easing bottlenecks. These higher-coarseness JTs are provided to XIMP alongside the other abstractions.

Formally, we define a simple resolution-lowering operator on a JT $T^{(r-1)} = (V^{(r-1)}, E^{(r-1)})$ with node features $f^{(r-1)}$, resolution $r$ and raw→reduced mapping $\mathcal{S}^{(r-1)}$. Nodes of degree one (leaves) are contracted into their unique parent, with all assignments in $\mathcal{S}^{(r-1)}$ redirected accordingly. Parent features are updated by a simple overwrite rule, $f^{(r)}(p) = \min f^{(r-1)}(p), f^{(r-1)}(\ell)$ for each leaf $\ell$, and duplicate mappings are removed. Edges incident to leaves are pruned, isolated nodes are preserved, and indices are compacted via an order-preserving relabeling $\phi : V^{(r-1)} \setminus L^{(r-1)} \rightarrow V^{(r)}$. The result is a reduced graph $T^{(r)} = (V^{(r)}, E^{(r)})$ with updated features $f^{(r)}$ and mapping $\mathcal{S}^{(r)}$, carrying forward the same number of raw nodes. We apply the resolution-lowering operator only to the JT, not the ErG, because (a) ErGs may contain substructures not reducible by this simple scheme (Figure 1), and (b) ErGs encode pharmacophoric properties absent in JTs, which focus solely on molecular structure (Section 2).

We show (under negligible assumptions) that the above scheme reduces inter-node communication distances in XIMP (proof in Appendix K), thereby potentially mitigating oversquashing and improving long-range interactions.

**Proposition 3.2.** *In XIMP's communication graph, contracting rings into single nodes and iteratively folding leaves into their parents with standard couplings never increases shortest-path distances. Distances between node pairs whose minimal path includes a subpath of at least three consecutive vertices inside a contracted region are strictly reduced. Successive rounds yield cumulative reductions.*

**Expressivity and Complexity.** While HIMP couples the molecular graph with a single junction-tree abstraction, XIMP supports multiple reduced graphs with both indirect and direct inter-message passing, strictly subsuming HIMP. XIMP admits richer representations via cross-abstraction embeddings, structured chemical priors, and multi-resolution coarsenings. We formalize this relation between HIMP and XIMP's expressivity as follows (proofs in Appendix I).

**Theorem 3.3.** *Let the hypothesis classes realized by HIMP and XIMP with depth $L$, hidden dimension $d_x$, and number of abstractions $n$ be denoted by $\mathcal{H}_{\mathrm{HIMP}}(L, d_x)$ and $\mathcal{H}_{\mathrm{XIMP}}(L, d_x, n)$. Then for any $L, d_x$ and $n \geq 1$, $\mathcal{H}_{\mathrm{HIMP}}(L, d_x) \subseteq \mathcal{H}_{\mathrm{XIMP}}(L, d_x, n)$.*

By construction, XIMP inherits and extends HIMP's ability to exceed 1-WL expressivity (Section 1). Moreover, to isolate the impact incorporating structurally different abstractions in conjunction has on expressivity (i.e., without introducing novel features), we show the following proposition.

**Proposition 3.4.** *There exist two molecules whose molecular graphs $G$, junction trees $T$, and extended reduced graphs $R$ are each indistinguishable by unlabeled 1-WL when considered in isolation, yet their compound graph $G^{+} = G \,\dot{\cup}\, T \,\dot{\cup}\, R$ (with the disjoint union $\dot{\cup}$) augmented with the inter-graph edges $E_{\mathrm{X}} = \{(v, u) \mid \mathbf{S}_T[v, u] = 1\} \cup \{(v, w) \mid \mathbf{S}_R[v, w] = 1\}$, is distinguishable by unlabeled 1-WL. Here $\mathbf{S}_T \in \{0, 1\}^{|V(G)| \times |V(T)|}$ and $\mathbf{S}_R \in \{0, 1\}^{|V(G)| \times |V(R)|}$ record atom-cluster (for $T$) and atom-ErG-node (for $R$) incidences.*

In our chemical setting, this relation corresponds to the I$^2$MP+MP communication graph in XIMP in the sense of Proposition 3.2. Furthermore, Proposition 3.4 suggests XIMP's expressivity benefits potentially transfer beyond molecular datasets, where abstractions or entire graphs may be unlabeled.

Regarding complexity, XIMP's per-layer runtime scales linearly with molecular graph size but adds a quadratic cost in hidden dimension and number of abstractions due to inter-message passing. Parameter count rises from HIMP's single quadratic term to additional linear and quadratic contributions, though modest in practice with few abstractions. Memory is dominated by node embeddings, with extra overhead from mapping matrices that also scale linearly and quadratically. Overall, XIMP maintains linear scaling with graph size and quadratic scaling with abstractions, which we consider a reasonable trade-off given the observed gains in predictive performance. For details, see Appendix H.

**Assumptions and Limitations.** In the chosen chemistry setting, XIMP assumes that abstractions such as junction trees (JT) and extended reduced graphs (ErG) provide useful inductive biases and that preprocessing (e.g., ring detection, pharmacophore tagging) yields accurate graph-abstraction mappings. Complexity grows polynomially in the number of abstractions $n$, so we restrict to $n \leq 3$ in practice. Empirically, evaluation is limited to MoleculeNet and Polaris; results may not generalize to all biochemical tasks. Hyperparameters were chosen by $k$-fold validation but tested with scaffold splits, potentially underestimating generalization. XIMP is strongest on ADMET tasks, while conventional GNNs remain competitive on properties driven by global structure.

| Model | ADMET ↓ | | | | | Potency ↓ | | MoleculeNet ↓ | | |
|---|---|---|---|---|---|---|---|---|---|---|
| | HLM | KSOL | LogD | MDR1-MDCKII | MLM | pIC50 MERS | pIC50 SARS | ESOL | FreeSolv | Lipo |
| ECFP | $0.56\,^{\pm}_{0.02}$ | $0.42\,^{\pm}_{0.01}$ | $0.86\,^{\pm}_{0.02}$ | $0.38\,^{\pm}_{0.01}$ | **$0.55\,^{\pm}_{0.02}$** | $0.79\,^{\pm}_{0.01}$ | $0.57\,^{\pm}_{0.01}$ | $1.21\,^{\pm}_{0.06}$ | $2.94\,^{\pm}_{0.06}$ | $0.73\,^{\pm}_{0.02}$ |
| GNN | $0.56\,^{\pm}_{0.03}$ | $0.48\,^{\pm}_{0.07}$ | **$0.68\,^{\pm}_{0.07}$** | $0.37\,^{\pm}_{0.02}$ | $0.64\,^{\pm}_{0.05}$ | $0.71\,^{\pm}_{0.02}$ | $0.45\,^{\pm}_{0.05}$ | **$0.71\,^{\pm}_{0.04}$** | **$1.58\,^{\pm}_{0.17}$** | **$0.52\,^{\pm}_{0.02}$** |
| HIMP | **$0.54\,^{\pm}_{0.05}$** | **$0.35\,^{\pm}_{0.04}$** | $0.80\,^{\pm}_{0.05}$ | **$0.35\,^{\pm}_{0.03}$** | $0.56\,^{\pm}_{0.06}$ | **$0.64\,^{\pm}_{0.03}$** | **$0.39\,^{\pm}_{0.04}$** | **$0.80\,^{\pm}_{0.07}$** | $1.77\,^{\pm}_{0.15}$ | **$0.52\,^{\pm}_{0.02}$** |
| XIMP | **$0.53\,^{\pm}_{0.08}$** | **$0.37\,^{\pm}_{0.04}$** | **$0.69\,^{\pm}_{0.03}$** | **$0.31\,^{\pm}_{0.03}$** | **$0.49\,^{\pm}_{0.02}$** | **$0.69\,^{\pm}_{0.05}$** | **$0.41\,^{\pm}_{0.03}$** | $0.82\,^{\pm}_{0.09}$ | $1.83\,^{\pm}_{0.17}$ | **$0.52\,^{\pm}_{0.02}$** |

Table 1: ADMET, Potency, and MoleculeNet results. Cells show 10-run mean/std of test MAE based on the hyperparameters that resulted in the lowest validation score. GNN abstracts GCN, GIN, GAT, GraphSAGE (i.e., best-performing GNN chosen as representative; fine-grained results in Appendix D). **Dark red bold** = best; **dark blue bold** = second-best (ties: all best are marked).

| Model | ADMET ↓ | | | | | Potency ↓ | | MoleculeNet ↓ | | |
|---|---|---|---|---|---|---|---|---|---|---|
| | HLM | KSOL | LogD | MDR1-MDCKII | MLM | pIC50 MERS | pIC50 SARS | ESOL | FreeSolv | Lipo |
| ECFP | **$0.48\,^{\pm}_{0.02}$** | $0.38\,^{\pm}_{0.02}$ | $0.72\,^{\pm}_{0.01}$ | $0.36\,^{\pm}_{0.02}$ | **$0.54\,^{\pm}_{0.02}$** | $0.75\,^{\pm}_{0.01}$ | $0.52\,^{\pm}_{0.02}$ | $1.16\,^{\pm}_{0.05}$ | $2.83\,^{\pm}_{0.07}$ | $0.73\,^{\pm}_{0.02}$ |
| GNN | $0.54\,^{\pm}_{0.05}$ | $0.45\,^{\pm}_{0.08}$ | **$0.67\,^{\pm}_{0.03}$** | $0.39\,^{\pm}_{0.03}$ | $0.56\,^{\pm}_{0.05}$ | $0.71\,^{\pm}_{0.02}$ | $0.42\,^{\pm}_{0.02}$ | **$0.71\,^{\pm}_{0.04}$** | **$1.62\,^{\pm}_{0.14}$** | **$0.52\,^{\pm}_{0.02}$** |
| HIMP | **$0.49\,^{\pm}_{0.04}$** | **$0.34\,^{\pm}_{0.06}$** | $0.81\,^{\pm}_{0.07}$ | **$0.33\,^{\pm}_{0.03}$** | $0.57\,^{\pm}_{0.06}$ | **$0.64\,^{\pm}_{0.03}$** | **$0.41\,^{\pm}_{0.06}$** | **$0.79\,^{\pm}_{0.07}$** | $1.82\,^{\pm}_{0.1}$ | $0.54\,^{\pm}_{0.02}$ |
| XIMP | **$0.48\,^{\pm}_{0.07}$** | **$0.33\,^{\pm}_{0.03}$** | **$0.69\,^{\pm}_{0.06}$** | **$0.32\,^{\pm}_{0.01}$** | **$0.52\,^{\pm}_{0.07}$** | **$0.68\,^{\pm}_{0.04}$** | **$0.38\,^{\pm}_{0.03}$** | $0.83\,^{\pm}_{0.08}$ | **$1.77\,^{\pm}_{0.16}$** | **$0.52\,^{\pm}_{0.01}$** |

Table 2: ADMET, Potency, and MoleculeNet results. Cells show 10-run mean/std test MAE for the best hyperparameters for each model chosen on the given test dataset. GNN abstracts GCN, GIN, GAT, GraphSAGE (i.e., best-performing GNN model shown; fine-grained results in Appendix D). **Dark red bold** = best; **dark blue bold** = second-best (ties: all best are marked).

## 4 EXPERIMENTAL EVALUATION

To evaluate the hypothesis that XIMP enhances predictive performance, we benchmarked it against HIMP, several widely used GNN baselines, and Extended Connectivity Fingerprints (ECFP) (Rogers & Hahn, 2010), a standard representation in molecular tasks. We briefly describe our experimental framework[1] and, to ensure reproducibility, provide details in Appendix A and C. For robust evidence, we conduct extensive hyperparameter search and ablations, totaling approximately 1M training runs.

**Models and Datasets.** Our setup consists of an encoder followed by a regression head. The regression head is implemented as a $k$-layer multilayer perceptron (MLP) with ReLU activations, applied to the graph-level embeddings produced by the encoder. For encoding, we compare GCN (Kipf & Welling, 2017), GIN (Li & Leskovec, 2022), GAT (Veličković et al., 2018), GraphSAGE (Hamilton et al., 2017), and HIMP (Section 2), plus an ECFP-based non-learnable baseline where only the regression head is trained. Together these cover convolutional, attention-based, and inductive aggregations. We evaluate on ten prediction tasks from MoleculeNet (Wu et al., 2018) and the Polaris challenge (ASAP Discovery x OpenADMET, 2025a;b), a recent dataset targeting ADMET (Absorption, Distribution, Metabolism, Excretion, Toxicity) endpoints and drug-candidate potency.

**Training, Hyperparameter Search.** We trained all models with Adam and MAE loss. Hyperparameters were chosen via stratified 10-fold cross-validation on binned regression targets to mitigate target imbalance during selection. We deliberately used target-based rather than purely scaffold-based CV: with small datasets (Appendix A), scaffold $k$-fold splits yield imbalanced target distributions and (at high $k$ with few scaffolds) small, uneven folds, making model selection unreliable. Hyperparameter tuning was performed via grid search. For final evaluation, we held out a $10\%$ *scaffold*-split test set to assess chemical-space generalization, retrained the selected configuration on the remaining data, and evaluated with MAE. We chose this protocol because, to our knowledge, no *stratified* scaffold split exists for regression: Joeres et al. (2025) consider classification only, and Zhang et al. (2025) control input-graph similarity rather than target stratification. This induces a mismatch –

---

[1] https://anonymous.4open.science/r/ximp-CF61/

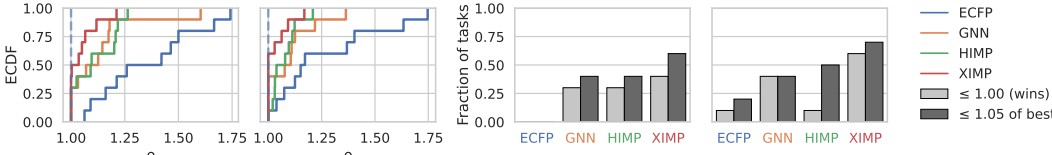

Figure 3: Performance profiles of architectures on ADMET, Potency, and MoleculeNet, computed from Tables 1 and 2. Panels 1-2 (left): Empirical Cumulative Distribution Functions (ECDFs) of the performance ratio $\rho = \mathrm{MAE}(\mathrm{model}, \mathrm{task})/\min_{\mathrm{arch}} \mathrm{MAE}(\mathrm{task})$, where $\rho = 1$ is best and $\rho > 1$ quantifies degradation; curves closer to the top-left indicate better overall performance. Panels 3-4 (right): discrete summaries across tasks. Bars show the fraction of tasks a model *wins* ($\rho = 1$) or is within a practical tolerance ($\rho \leq \tau$, here $\tau = 1.05$). Panels 1, 3 use test MAE with hyperparameters chosen by validation; panels 2, 4 use test MAE under best hyperparameters per model and dataset.

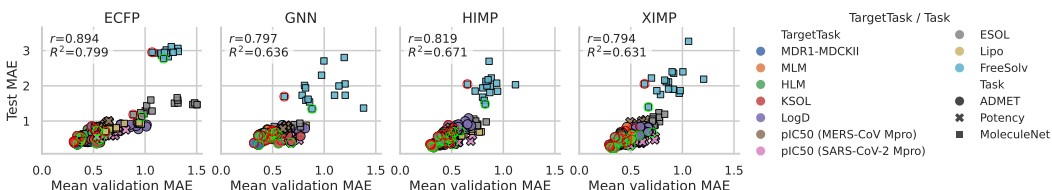

Figure 4: Validation MAE (x-axis) versus test MAE (y-axis) across hyperparameter configurations. For each task, the lowest test MAE is marked by a green circle and the lowest validation MAE by a red circle. Each panel reports the Pearson correlation ($r$) and $R^2$ score. Points closer to the lower left indicate better performance. To improve readability, non-optimal runs (neither red nor green) were randomly subsampled at $150$ per architecture.

stratified CV for selection vs. scaffold-based testing – that can complicate hyperparameter choice, as validation folds might not match the test distribution; we nonetheless adopt this conservative setup because scaffold splitting yields a structurally distinct and more realistic test set. Accordingly, we report both, mean/std of the test MAE for hyperparameters selected via mean validation MAE across stratified folds (Table 1) for final assessment, and mean/std of the test MAE for optimal test-selected hyperparameters on the scaffold-split holdout (Table 2) as an optimistic upper bound on performance to illustrate what could be achieved with an ideal hyperparameter search.

**Results.** We find that XIMP yields the best predictive performance for the hyperparameters chosen via mean validation MAE in $4/10$ cases, whereas the strongest baseline method (HIMP) likewise outperforms it's competitors on a given dataset in $4/10$ cases (Table 1). We note that choosing our model according to this criterion yields less than optimal results, see Figure 4. This stems from a mismatch between the assumed distributions of target and scaffolds in the training and test data.

Generally though, we find that XIMP is the most robust of the evaluated methods. This is reflected in the aggregate view we provide in Figure 3 (left), which shows the per-architecture ECDFs of the results of XIMP and its baselines and is equivalent to the performance profiles introduced by Dolan & Moré (2002), a standard method for benchmarking algorithms. XIMP's performance profile dominates all baselines, both for validation selected hyperparameters as well as globally best hyperparameters, underlining its robustness across diverse tasks. Figure 3 (right), which provides a discrete summary perspective across tasks, further underlines XIMP's robustness: more often than any of it's baselines, XIMP is either the best performing model measured by it's performance ratio $\rho$ or within a practical tolerance of the best performing model.

On a more granular level, Tables 1 and 2 both further confirm the strength of XIMP's ability to integrate multiple chemically meaningful and interpretable reduced graphs (junction trees for coarse-grained connectivity, ErGs for fine-grained and pharmacophoric patterns) with the molecular graph itself. This gives it an inductive bias toward functional groups, pharmacophores, and scaffold structures, and, unlike other methods, XIMP also allows for direct message passing between these abstractions, allowing new features to be learned in ways not previously possible. Hence, as expected, for the ADMET tasks that take advantage of pharmacophoric features (such as HLM, MDR1-MDCKII,

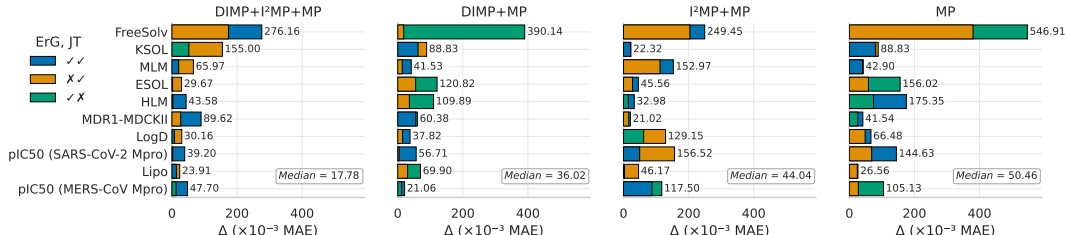

Figure 5: Relative test error across ErG/JT settings per architecture with medians across targets. Each panel shows $\Delta$ in milli-MAE by target task (x-axis), colored by whether ErG, JT or a combination is used. Here $\Delta = 1000 \times (\text{MAE} - \min_{\text{cfg}} \text{MAE})$, so the best setting is $0$; the other bars report the gap. Bar labels report exact $\Delta$ values, and a horizontal line at $0$ marks the per-target optimum.

MLM), XIMP typically performs better than its competitors, due to its ability to include chemically relevant features (H-Acceptor, H-Donor). Conversely, for tasks that rely on the overall structure of the molecule (ESOL, FreeSolv, Lipophilicity), conventional GNNs perform competitively.

In low-data regimes, the inductive bias of XIMP also provides a clear advantage. For example, the ADMET dataset comprises only $560$ molecules, in contrast to the substantially larger Potency and MoleculeNet datasets (see Appendix A). Under these conditions, XIMP achieves superior performance in $3/5$ or $4/5$ tasks respectively, outperforming competing methods (Tables 1 and 2).

**Hyperparameters and Ablation.** We also observe that there is a large discrepancy between test MAE for the hyperparameters chosen via mean validation MAE and globally best test MAE (Figure 4). For the globally best test MAE, we find that XIMP outperforms its baselines in $6/10$ cases and specifically on $4/5$ ADMET tasks, whereas the best performing baseline methods (GNNs) only accomplish this in $4/10$ cases (the best-performing specific GNN is GIN, which is best in $2/10$ cases, see Appendix D). While evaluating hyperparameters optimally chosen on the test set naturally constitutes data leakage, we include these results as an optimistic upper bound to performance, indicating the possibility for improvement under perfect tuning.

Finally, we evaluated how well the components of XIMP exploit multiple graph abstractions (i.e., ERG and JT). DIMP+MP is typically the best performing configuration for multi-abstraction settings where ERG and JT are both active (Figure 5). In settings where I²MP+MP or I²MP+DIMP+MP are used, XIMP shows a preference for the ERG abstraction. For the configuration without DIMP or I²MP, no clear trend emerges; the utility of abstraction combinations appears largely dataset- and task-dependent. The median relative test error across targets indicates that I²MP+DIMP is most robust to abstraction choice, whereas MP alone is least, underscoring XIMP's adaptability to diverse abstraction combinations. A more detailed ablation study is provided in Appendix E.

## 5 CONCLUSION

We presented XIMP, a versatile inter-message-passing framework that learns over any number of arbitrary graph abstractions within a single model. XIMP enables both indirect (graph $\leftrightarrow$ abstractions) and direct (abstraction $\leftrightarrow$ abstraction) communication, a learned multi-view readout, uses abstractions of varying *coarseness* to mitigate oversquashing and improve long-range communication. Although domain-agnostic, we instantiate it in a chemistry setting with junction trees and extended reduced graphs to demonstrate how interpretable abstractions can be exploited. Across ten diverse property-prediction tasks, XIMP matches or surpasses strong GNN baselines and fixed fingerprints, with gains attributable to explicit cross-abstraction messaging. Our results position multi-abstraction message passing as a principled approach for data-scarce regimes and interpretable graph learning.

Future work includes extending XIMP to additional reductions and attribution tools for tracing predictions across abstractions and extending XIMP to protein design via multi-level graph reductions (atoms$\rightarrow$residues$\rightarrow$primary/secondary/tertiary structures). Such hierarchical integration may uncover new insights in large protein graphs. Moreover, considering discrepancy for different hyperparameter choices, we aim to research stratified-scaffold cross-validation methods for regression to make hyperparameter search more reliable and consistent in chemistry settings.

**Ethics Statement.** This work introduces a novel graph neural network architecture and demonstrates its advantages in molecular property prediction. All experiments use publicly available datasets (e.g., MoleculeNet, Polaris), released under appropriate licenses, with no human or personally identifiable data involved.

We recognize the potential for both positive and negative societal impact. On the positive side, improved molecular property prediction can accelerate drug discovery and related biomedical research. On the negative side, similar techniques could in principle be misused to aid the design of harmful compounds. Our work does not perform molecule generation, and we strongly caution against unsafe or unregulated deployment of predictive models in high-stakes applications.

From an environmental perspective, we limited training runs and hyperparameter searches to scales comparable with prior molecular GNN studies. To further mitigate computational waste, we release code, configuration files, and preprocessing scripts to ensure full reproducibility.

Finally, as with all data-driven methods, our models may inherit biases from the underlying datasets, for example underrepresentation of certain chemical classes. We encourage practitioners to evaluate these limitations before applying such models in real-world discovery settings.

**Reproducibility Statement.** To facilitate reproducibility, we provide an anonymized code repository (`https://anonymous.4open.science/r/ximp-CF61/`) that includes all datasets required for training and evaluation, removing the need to download from third-party sources. The repository contains a detailed README with step-by-step instructions to replicate the main experiments, and proofs supporting our theoretical claims are included in the Appendix. All experiments were conducted on our group's local compute cluster, with hardware specifications reported in Appendix A. The total compute time, including hyperparameter search and evaluation, amounted to approximately five weeks.

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

| GPUs | CPU | Memory |
|---|---|---|
| 8× H100 | Xeon Platinum 8480C (224 thr) | GPU 640 GB;  RAM ∼2 TB |
| 8× V100 16 GB | Xeon E5−2698 (80 thr) | GPU 128 GB;  RAM ∼500 GB |
| 4× H100 | Xeon Platinum 8468V (192 thr) | GPU ∼374 GiB;  RAM ∼1 TB |

Table 3: Specifications of GPU compute nodes (node names omitted).

| Dataset Source | Targets | | Molecules |
|---|---|---|---|
| | Property | Endpoint | |
| MoleculeNet | Solubility | ESOL | 1,128 |
| | Hydration Free Energy | FreeSolv | 642 |
| | Lipophilicity | Lipo | 4,200 |
| Polaris ADMET | Metabolic Stability (Human) | HLM | |
| | Metabolic Stability (Mouse) | MLM | |
| | Solubility | KSOL | 560 |
| | Distribution Coefficient | LogD | |
| | Membrane Permeability | MDR1-MDCKII | |
| Polaris Potency | MERS-CoV Mpro | pIC50 | |
| | SARS-CoV-2 Mpro | pIC50 | 1,328 |

Table 4: Summary of datasets MoleculeNet (Wu et al., 2018), Polaris ADMET (ASAP Discovery x OpenADMET, 2025a) and Polaris Potency (ASAP Discovery x OpenADMET, 2025b) used for molecular property and potency prediction.

## A    DATASETS & HARDWARE

**Datasets**    We evaluated our models on molecular property prediction tasks from two dataset collections: MoleculeNet (Wu et al., 2018) and Polaris (ASAP Discovery x OpenADMET, 2025a;b). From MoleculeNet, we selected three standard regression benchmarks: solubility (ESOL), hydration free energy (FreeSolv), and lipophilicity (Lipo). From the Polaris ADMET dataset, we considered four targets relevant to drug absorption and distribution: human liver microsomal stability (HLM), mouse liver microsomal stability (MLM), solubility (KSOL), membrane permeability (MDR1-MDCKII), and lipophilicity (LogD). In addition, we included two potency prediction tasks from Polaris, specifically targeting drug candidate activity against the main proteases of MERS-CoV (MERS-CoV-Mpro) and SARS-CoV-2 (SARS-CoV-2-Mpro), using the pIC50 endpoint. Please find dataset statistics in Table 4 and hardware specifications in Table 3.

## B    REDUCED GRAPHS CONSTRUCTION

**Junction Trees**    To construct a *junction tree*, the process begins by converting the molecular structure into a graph. This graph is then partitioned into substructures by identifying all simple rings using RDKit's `GetSymmSSSR` function (rdk, 2025). Next, all edges not belonging to any cycle are identified. Each simple ring and each such edge is treated as a separate cluster, represented as a node in the cluster graph, where a cluster consists of the set of atoms in the corresponding ring, or the two atoms connected by the corresponding edge. Edges are added between clusters if they share at least one atom.

The resulting cluster graph may itself contain cycles, which can lead to non-unique mappings from the molecular graph to the cluster graph. This can occur, for example, when an atom has three or more substituents. To prevent this, the intersecting atom is added as its own cluster, and the bonds responsible for the cycle are removed. Eliminating these cycles ensures an injective mapping from the molecular graph to the cluster graph and removes ambiguities in the decomposition, yielding a unique and well-defined representation. After cycle removal, the junction tree is obtained. The full process is illustrated in Figure 1a.

For each node in the junction tree, the following categories are assigned as one-hot encodings: {singleton, bond, ring, bridged compound}. The edges do not contain any feature information. From a chemical perspective, this decomposition facilitates information flow across multiple fused rings in graph neural networks. For instance, in conventional message passing, the top left carbon atom in the aromatic ring shown in Figure 1a could not exchange information with the chlorine atom because they are five hops apart. In the junction tree, however, they are only two hops apart. This condensation of cyclic structures enables the model to capture longer-range interactions more efficiently.

**Extended reduced Graphs.** An *extended reduced graph* (ErG) (Stiefl et al., 2006) is a simplified molecular representation that captures pharmacophoric and interaction-relevant features while abstracting unnecessary atomic detail. It builds on reduced graphs, which represent chemically meaningful moieties (e.g., H-bond donors/acceptors, aromatic rings) as pseudo-nodes linked according to relationships in the parent molecule. ErGs extend this by treating ring systems separately from hydrogen-bonding and charge features—where prior methods often folded these into the ring node—improving discrimination between compounds with different molecular skeletons. They also fully enumerate "flip-flop" atoms that can act as either H-bond donors or acceptors.

The process of generating an ErG from a chemical structure, as well as the node mapping is shown described below and shown in Figure 1b.

1. Atoms are **charged** to represent the molecule under physiological conditions.

2. **Initial Atom Property Assignment**: H-bond donor and H-bond acceptor properties are assigned to the atoms. Atoms that can function as both H-bond donor and H-bond acceptor receive a distinct *flip-flop* property.

3. **Endcap Group Identification**: These are lateral hydrophobic features, typically composed of three atoms. Thioethers are also identified as endcap groups, and are assigned a property.

4. **Ring System Abstraction**: Rings are abstracted as follows to capture their overall properties:

   (a) Add a centroid atom for each ring and assign it a feature (*aromatic, hydrophobic*).
   (b) Retain all substituted ring atoms and create bonds from the centroid to that atom.
   (c) Retain all bridgehead atoms (atoms belonging to two or more rings), and create bonds from the centroids to those atoms.
   (d) Remove all non-substituted ring atoms and retain all bonds between the atoms that were retained in the previous two steps.

During the ring abstraction process, if any atom constituting the ring is assigned a specific property, that atom is considered a connected node within the ring framework. The ErG encodes the following node features: *H-bond donor, H-bond acceptor, positive charge, negative charge, hydrophic, aromatic*.

## C  HYPERPARAMETER SEARCH SPACE

For the GNN-based methods, we explored the following hyperparameter space: number of message-passing layers (1, 2, or 3), hidden dimensions (16 or 32), output embedding dimension (16 or 32), batch size (64 or 128), hidden dimension of the regression head (16 or 32), and number of training epochs (50, 100, or 150). The learning rate, Adam weight decay, dropout rate, and the number of bins for regression stratification remained fixed at $10^{-3}$, $10^{-4}$, 0.1, and 10, respectively.

For XIMP specifically, we included additional hyperparameters controlling the selection of graph abstraction schemes (ERG vs. JT), the choice of message-passing schemes ($I^2$MP vs. DIMP (Section 3), binary hyperparameters), and the granularity (resolution) of JT, varied as an integer from 1 to 3. Moreover, we included the selection of the embedding dimension of the reduced graphs (16 and 32). For HIMP, we included a binary hyperparameter choosing whether inter-message passing is active or not.

For ECFP, we examined the daylight atomic invariants initial atom identifiers with output channels (16, 32, 1024, 2048). We decided to include the fingerprints of size 16 and 32 for completeness and to allow for a direct comparison with GNN model embedding dimensions. The fingerprint radius was set to values (2, 3, or 4) resulting in ECFP_4, ECFP_6, and ECFP_8 variants.

| Model | ADMET ↓ | | | | | Potency ↓ | | MoleculeNet ↓ | | |
|---|---|---|---|---|---|---|---|---|---|---|
| | HLM | KSOL | LogD | MDR1-MDCKII | MLM | pIC50 MERS | pIC50 SARS | ESOL | FreeSolv | Lipo |
| ECFP | $0.56_{\pm0.02}$ | $0.42_{\pm0.01}$ | $0.86_{\pm0.02}$ | $0.38_{\pm0.01}$ | $0.55_{\pm0.02}$ | $0.79_{\pm0.01}$ | $0.57_{\pm0.01}$ | $1.21_{\pm0.06}$ | $2.94_{\pm0.06}$ | $0.73_{\pm0.02}$ |
| GAT | $0.62_{\pm0.03}$ | $0.52_{\pm0.05}$ | $0.75_{\pm0.08}$ | $0.41_{\pm0.02}$ | $0.66_{\pm0.05}$ | $0.71_{\pm0.02}$ | $0.45_{\pm0.05}$ | $0.74_{\pm0.04}$ | $2.02_{\pm0.32}$ | $0.61_{\pm0.02}$ |
| GCN | $0.57_{\pm0.05}$ | $0.48_{\pm0.07}$ | $\mathbf{0.68}_{\pm0.07}$ | $0.40_{\pm0.04}$ | $0.66_{\pm0.04}$ | $0.71_{\pm0.02}$ | $0.48_{\pm0.05}$ | $0.74_{\pm0.02}$ | $1.75_{\pm0.25}$ | $0.59_{\pm0.02}$ |
| GIN | $0.56_{\pm0.02}$ | $0.50_{\pm0.10}$ | $0.75_{\pm0.06}$ | $0.41_{\pm0.06}$ | $0.64_{\pm0.05}$ | $0.73_{\pm0.02}$ | $0.46_{\pm0.04}$ | $\mathbf{0.71}_{\pm0.04}$ | $1.91_{\pm0.27}$ | $\mathbf{0.52}_{\pm0.02}$ |
| GraphSAGE | $0.56_{\pm0.03}$ | $0.49_{\pm0.07}$ | $0.69_{\pm0.06}$ | $0.37_{\pm0.02}$ | $0.64_{\pm0.05}$ | $0.73_{\pm0.02}$ | $0.52_{\pm0.05}$ | $0.73_{\pm0.04}$ | $\mathbf{1.58}_{\pm0.17}$ | $0.54_{\pm0.02}$ |
| HIMP | $0.54_{\pm0.05}$ | $\mathbf{0.35}_{\pm0.04}$ | $0.80_{\pm0.05}$ | $0.35_{\pm0.03}$ | $0.56_{\pm0.06}$ | $\mathbf{0.64}_{\pm0.03}$ | $\mathbf{0.39}_{\pm0.04}$ | $0.80_{\pm0.07}$ | $1.77_{\pm0.15}$ | $\mathbf{0.52}_{\pm0.02}$ |
| XIMP | $\mathbf{0.53}_{\pm0.08}$ | $0.37_{\pm0.04}$ | $0.69_{\pm0.03}$ | $\mathbf{0.31}_{\pm0.03}$ | $\mathbf{0.49}_{\pm0.02}$ | $0.69_{\pm0.05}$ | $0.41_{\pm0.03}$ | $0.82_{\pm0.09}$ | $1.83_{\pm0.17}$ | $\mathbf{0.52}_{\pm0.02}$ |

Table 5: Potency, ADMET, and MoleculeNet results. Cells show the test MAE based on the hyperparameters that resulted in the lowest validation score. The best result for a given task is marked in bold.

| Model | ADMET ↓ | | | | | Potency ↓ | | MoleculeNet ↓ | | |
|---|---|---|---|---|---|---|---|---|---|---|
| | HLM | KSOL | LogD | MDR1-MDCKII | MLM | pIC50 MERS | pIC50 SARS | ESOL | FreeSolv | Lipo |
| ECFP | $\mathbf{0.48}_{\pm0.02}$ | $0.38_{\pm0.02}$ | $0.72_{\pm0.01}$ | $0.36_{\pm0.02}$ | $0.54_{\pm0.02}$ | $0.75_{\pm0.01}$ | $0.52_{\pm0.02}$ | $1.16_{\pm0.05}$ | $2.83_{\pm0.07}$ | $0.73_{\pm0.02}$ |
| GAT | $0.59_{\pm0.04}$ | $0.45_{\pm0.08}$ | $0.76_{\pm0.05}$ | $0.40_{\pm0.05}$ | $0.66_{\pm0.07}$ | $0.71_{\pm0.02}$ | $0.42_{\pm0.02}$ | $0.74_{\pm0.05}$ | $1.82_{\pm0.27}$ | $0.60_{\pm0.02}$ |
| GCN | $0.54_{\pm0.05}$ | $0.50_{\pm0.07}$ | $0.69_{\pm0.05}$ | $0.39_{\pm0.05}$ | $0.64_{\pm0.06}$ | $0.71_{\pm0.02}$ | $0.45_{\pm0.03}$ | $0.75_{\pm0.02}$ | $\mathbf{1.62}_{\pm0.14}$ | $0.59_{\pm0.03}$ |
| GIN | $0.54_{\pm0.03}$ | $0.50_{\pm0.07}$ | $0.70_{\pm0.04}$ | $0.40_{\pm0.06}$ | $0.56_{\pm0.05}$ | $0.74_{\pm0.01}$ | $0.49_{\pm0.07}$ | $\mathbf{0.71}_{\pm0.04}$ | $2.02_{\pm0.33}$ | $\mathbf{0.52}_{\pm0.02}$ |
| GraphSAGE | $0.56_{\pm0.04}$ | $0.52_{\pm0.08}$ | $\mathbf{0.67}_{\pm0.03}$ | $0.39_{\pm0.03}$ | $0.63_{\pm0.03}$ | $0.73_{\pm0.03}$ | $0.50_{\pm0.07}$ | $0.73_{\pm0.04}$ | $2.06_{\pm0.54}$ | $0.54_{\pm0.02}$ |
| HIMP | $0.49_{\pm0.04}$ | $0.34_{\pm0.06}$ | $0.81_{\pm0.07}$ | $0.33_{\pm0.03}$ | $0.57_{\pm0.06}$ | $\mathbf{0.64}_{\pm0.03}$ | $0.41_{\pm0.06}$ | $0.79_{\pm0.07}$ | $1.82_{\pm0.10}$ | $0.54_{\pm0.02}$ |
| XIMP | $\mathbf{0.48}_{\pm0.07}$ | $\mathbf{0.33}_{\pm0.03}$ | $0.69_{\pm0.06}$ | $\mathbf{0.32}_{\pm0.01}$ | $\mathbf{0.52}_{\pm0.07}$ | $0.68_{\pm0.04}$ | $\mathbf{0.38}_{\pm0.03}$ | $0.83_{\pm0.08}$ | $1.77_{\pm0.16}$ | $\mathbf{0.52}_{\pm0.01}$ |

Table 6: Potency, ADMET, and MoleculeNet results. Cells show test MAE for the best hyperparameters for each model chosen on the given dataset. The best result for a given task is marked in bold.

## D    EXTENDED RESULTS

In Tables 5 and 6, we provide a fine-grained view of our results that more clearly highlights the limitations of conventional GNNs than the aggregated summary in the main paper. Across both tables, conventional architectures (GIN, GCN, GAT, GraphSAGE) are the top performer on at most 2/10 tasks, whereas XIMP leads on 4/10 tasks with validation-selected hyperparameters and on 6/10 tasks with per-dataset best hyperparameters (evaluated on the test set). This underscores the benefit of advanced inter-message-passing methods that integrate chemically meaningful, interpretable graph abstractions: XIMP, in particular, shows a clear advantage over competing architectures, especially under the best-hyperparameter setting.

## E    ABLATION STUDY

We conducted an extensive evaluation study to evaluate the impact of the different message passing schemes ($I^2$MP and DIMP), graph reductions (ERG, JT), and their resolutions. As we deem it the most relevant metric due to the necessity to generalize to chemical scaffold unseen during training, we conduct our ablation study using test MAE of the models trained with the hyperparameters chosen via mean validation MAE.

**Message Passing Schemes**    Concerning message passing schemes, we find that for the test MAE of the models trained with the hyperparameters chosen via mean validation MAE, DIMP yields the best results most reliably. As shown in Table 7, XIMP employing DIMP outperforms other higher order message passing schemes in only 5 out of 10 cases, which is seconded by XIMP with DIMP+$I^2$MP. Our results together illustrate the effectiveness of our devised higher order intra message passing schemes and likewise highlight the necessity of careful hyperparameter selection.

| Model | ADMET ↓ | | | | | Potency ↓ | | MoleculeNet ↓ | | |
|---|---|---|---|---|---|---|---|---|---|---|
| | HLM | KSOL | LogD | MDR1-MDCKII | MLM | pIC50 MERS-CoV | pIC50 SARS-CoV-2 | ESOL | FreeSolv | Lipo |
| XIMP (a) | 0.543 | 0.382 | 0.731 | **0.319** | 0.625 | **0.644** | 0.404 | 0.794 | 1.907 | 0.526 |
| XIMP (b) | **0.486** | **0.280** | **0.726** | 0.365 | 0.507 | **0.644** | **0.371** | 0.794 | 2.054 | **0.494** |
| XIMP (c) | **0.486** | **0.280** | 0.754 | 0.330 | **0.493** | 0.657 | 0.515 | 0.818 | 2.054 | 0.512 |
| XIMP (d) | 0.569 | 0.297 | 0.801 | 0.376 | **0.493** | 0.761 | 0.404 | **0.765** | **1.877** | 0.536 |

Table 7: XIMP ablation (a/b/c/d) with XIMP (a) ($I^2$MP+MP), XIMP (b) (DIMP+MP), XIMP (c) (MP; plain message passing with late fusion of the different graph-level embedings) across all target tasks (columns), and XIMP (d) (DIMP+$I^2$MP+MP). Test MAE of validation-selected configs only. The best result for a task is marked in bold.

| Model | JT coar. | ADMET ↓ | | | | | Potency ↓ | | MoleculeNet ↓ | | |
|---|---|---|---|---|---|---|---|---|---|---|---|
| | | HLM | KSOL | LogD | MDR1-MDCKII | MLM | pIC50 MERS-CoV | pIC50 SARS-CoV-2 | ESOL | FreeSolv | Lipo |
| XIMP (a) | 1 | **0.483** | **0.337** | 0.757 | **0.298** | **0.513** | 0.812 | **0.453** | **0.812** | 1.952 | 0.526 |
| XIMP (a) | 2 | 0.543 | 0.387 | 0.731 | 0.315 | 0.625 | **0.731** | 0.501 | 0.920 | 2.043 | **0.521** |
| XIMP (a) | 3 | 0.555 | 0.382 | **0.730** | 0.359 | 0.529 | 0.795 | 0.542 | 0.890 | **1.800** | 0.583 |
| XIMP (b) | 1 | **0.450** | **0.343** | 0.726 | **0.350** | 0.535 | **0.665** | 0.428 | **0.794** | **1.664** | **0.494** |
| XIMP (b) | 2 | 0.593 | 0.452 | **0.698** | 0.360 | **0.488** | 0.852 | **0.363** | 0.825 | 1.743 | 0.527 |
| XIMP (b) | 3 | 0.548 | 0.386 | 0.731 | 0.369 | 0.489 | 0.689 | 0.397 | 0.828 | 2.080 | 0.529 |
| XIMP (c) | 1 | 0.536 | 0.361 | 0.792 | **0.315** | **0.491** | **0.630** | 0.515 | **0.759** | 1.890 | **0.513** |
| XIMP (c) | 2 | **0.512** | 0.368 | 0.754 | 0.364 | 0.516 | 0.709 | **0.403** | 0.816 | **1.507** | 0.538 |
| XIMP (c) | 3 | 0.661 | **0.321** | **0.683** | 0.346 | 0.639 | 0.707 | 0.434 | 0.889 | 1.796 | 0.516 |
| XIMP | 1 | **0.536** | 0.333 | 0.829 | 0.376 | **0.493** | **0.796** | 0.440 | 0.765 | 1.979 | 0.536 |
| XIMP | 2 | 0.569 | **0.297** | 0.801 | 0.343 | 0.517 | 0.848 | **0.409** | **0.749** | **1.862** | **0.530** |
| XIMP | 3 | 0.595 | 0.376 | **0.728** | **0.342** | 0.557 | 0.958 | 0.441 | 0.935 | 2.138 | 0.550 |

Table 8: XIMP ablation (a/b/c/base) with XIMP (a) ($I^2$MP only), XIMP (b) (DIMP only), and XIMP (c) (neither; plain message passing with late fusion of the different graph-level embedings) across all target tasks (columns) showing the test MAE for configurations selected by the lowest validation MAE within each jt_coarseness (rows) and target task (columns). The table isolates how changing jt_resolution $\in \{1, 2, 3\}$ impacts generalization for each XIMP variant across targets. Lower is better. The absolute per-target minima (best raw test MAE) for each variant and task are marked in bold.

**Junction Tree Resolution.** Moreover, we evaluated how XIMP with different combinations of intermessage passing (i.e., $I^2$MP, DIMP) responds to different JT resolutions (Table 8). To this purpose, we conducted the following experiments only using only JTs and evaluated test MAE for the hyperparameters selected by lowest validation MAE per jt_resolution. Our results indicate that this is in large parts a datasets and task dependent property. For example, in ADMET's KSOL and LogD tasks, a resolution of 2 or more appeared to be preferable in most DIMP and $I^2$MP combinations, whereas, for example, for the MDR1-MDCKII and MLM tasks, a resolution of 1 was preferable in almost all cases. For the pIC50 tasks as well as ESOL and FreeSolv, the architectural combination seemed to have a larger impact; with, for example a resolution of 1 being optimal for pIC50 (SARS-CoV2 Mpro) and XIMP with $I^2$MP and a resolution of 3 being optimal for pIC50 (SARS-CoV2 Mpro) and XIMP with DIMP. To conclude, it appears that the optimal combination of message passing schemes and feature tree resolution is not only highly dependent on the underlying data but also the task to be learned; indicating latent mechanisms deeply rooted in the chemical relevance of the abstractions and the corresponding information flow between them.

## F  GRAPH ISOMORPHISM NETWORKS

The GIN update (matrix notation, used henceforth) is

$$\boldsymbol{X}^{(l+1)} \;=\; \mathrm{MLP}^{(l)}\Big(\big(\boldsymbol{A} + (1 + \epsilon^{(l)})\,\boldsymbol{I}\big)\,\boldsymbol{X}^{(l)}\Big),$$

where $\boldsymbol{A} \in \{0,1\}^{|V(G)| \times |V(G)|}$ denotes the adjacency matrix, $\boldsymbol{I} \in \mathbb{R}^{|V(G)| \times |V(G)|}$ the identity, $\boldsymbol{X}^{(l)} \in \mathbb{R}^{|V(G)| \times d_x}$ the node feature matrix at layer $l$, and $\epsilon^{(l)} \in \mathbb{R}$ a learnable scalar. For GIN-E the layer update is given by

$$\boldsymbol{X}^{(l+1)} \;=\; \mathrm{MLP}^{(l)}\Big(\big(1 + \epsilon^{(l)}\big)\,\boldsymbol{X}^{(l)} \;+\; \mathcal{M}_{\boldsymbol{A}}^{(l)}(\boldsymbol{X}^{(l)}, \boldsymbol{E})\Big)$$

with

$$[\mathcal{M}_{\boldsymbol{A}}^{(l)}(\boldsymbol{X}, \boldsymbol{E})]_v \;=\; \sum_{u=1}^{n} A_{vu}\,\sigma\Big(\boldsymbol{x}_u^{(l)} \;+\; \boldsymbol{E}_{vu}\Big),$$

where $\boldsymbol{E} \in R^{|V(G)| \times |V(G)| \times d_e}$ denotes the tensor of edge feature vectors and $\boldsymbol{E}_{vu}$ the edge feature vector for the edge between nodes $u$ and $v$. If $d_e \neq d_x$, an additional learnable transformation can be applied to $\boldsymbol{E}_{vu}$ to transform the edge feature vector to the space of the node embeddings.

## G  HIERACHICAL INTER-MESSAGE PASSING (HIMP)

In the case where the abstracted graph is hierarchical (such as a JT), HIMP follows the follow scheme:

Let $\boldsymbol{X}^{(l)} \in \mathbb{R}^{|V(G)| \times d_x}$ and $\boldsymbol{T}^{(l)} \in \mathbb{R}^{|V(T)| \times d_x}$ denote matrices storing the embeddings of nodes of $G$ and $T$ in layer $l$, respectively and $\boldsymbol{S} \in \{0,1\}^{|V(G)| \times |V(T)|}$ be the mapping matrix that captures the assignment of nodes of the molecular graph to nodes of the JT. Then the inter-message passing step changes the embedding matrices $\boldsymbol{X}^{(l)}$ and $\boldsymbol{T}^{(l)}$ according to

$$\boldsymbol{X}^{(l)} \leftarrow \boldsymbol{X}^{(l)} + \sigma\Big(\boldsymbol{S}\boldsymbol{T}^{(l)}\boldsymbol{W}_1^{(l)}\Big)$$

$$\boldsymbol{T}^{(l)} \leftarrow \boldsymbol{T}^{(l)} + \sigma(\boldsymbol{S}^T \boldsymbol{X}^{(l+1)} \boldsymbol{W}_2^{(l)}),$$

where $\sigma$ denotes a non-linearity and the matrices $\boldsymbol{W}_1^{(l)}$ and $\boldsymbol{W}_2^{(l)} \in \mathbb{R}^{d_x \times d_x}$ are trainable parameters specific for layer $l$.

The READ function after layer $L$ is given by

$$\boldsymbol{h}_G = \frac{1}{|V(G)|} \sum_{i=1}^{|V(G)|} \boldsymbol{x}_i^{(L)} \bigoplus \frac{1}{|V(T)|} \sum_{i=1}^{|V(T)|} \boldsymbol{t}_i^{(L)},$$

where $\bigoplus$ denotes an aggregation operation (i.e., concatenation $||$ or summation $\sum$), $\boldsymbol{x}_i^{(L)} \in \mathbb{R}^{d_x}$ and $\boldsymbol{t}_i^{(L)} \in \mathbb{R}^{d_x}$ are the final embeddings of the $i$-th node of the graph $G$ and tree $T$, respectively.

## H  COMPLEXITY ANALYSIS

Below, we provide an analysis of the time and space complexity of XIMP.

**Per-layer time complexity.**  For HIMP, each message-passing layer consists of (i) intra-graph updates on the molecular graph $G$ with $|V(G)|$ nodes and adjacency $\boldsymbol{A}$, and (ii) updates on its junction tree $T$ with $|V(T)|$ nodes. This yields a per-layer cost of $\mathcal{O}(|E(G)|d_x + |V(G)|d_x^2 + |V(T)|d_x^2)$, dominated by aggregation and MLP operations. XIMP generalizes this to $n$ reduced graphs $T_1, \ldots, T_n$, adding (a) indirect inter-message passing ($G \leftrightarrow T_i$) at cost $\mathcal{O}\big(\sum_{i=1}^{n} |V(G)|d_x^2\big)$ and (b) direct inter-message passing ($T_i \leftrightarrow T_j$ for $i < j$) at cost $\mathcal{O}\big(\sum_{i<j} |V(T_i)|d_x^2 + |V(T_j)|d_x^2\big)$. Thus, per-layer complexity scales as

$$\mathcal{O}\Big(|E(G)|d_x + \big(|V(G)| + \textstyle\sum_i |V(T_i)|\big)d_x^2 + n|V(G)|d_x^2 + \textstyle\sum_{i<j}\big(|V(T_i)| + |V(T_j)|\big)d_x^2\Big),$$

which is polynomial in the number of abstractions $n$. Compared to HIMP, XIMP introduces only moderate quadratic overhead in $d_x$, while enabling richer cross-abstraction communication.

**Parameter count.** HIMP maintains two GNN encoders (GIN-E on $G$, GIN on $T$) plus linear projections for inter-message passing. The parameter count therefore scales as $\Theta(L \cdot d_x^2)$, with constants depending on MLP depth. XIMP extends this by (i) duplicating the abstraction encoder for each $T_i$, and (ii) introducing additional projection matrices $\mathbf{W}_{i,1}, \mathbf{W}_{i,2} \in \mathbb{R}^{d_x \times d_x}$ for indirect inter-message passing and $\mathbf{W}_{i \to j} \in \mathbb{R}^{d_x \times d_x}$ for direct inter-abstraction exchange. The resulting parameter count is

$$\Theta\Big(L \cdot ((1+n)\,d_x^2 + n \cdot d_x^2 + n^2 \cdot d_x^2)\Big),$$

dominated by $\mathcal{O}(n^2 d_x^2)$ for pairwise abstractions. While this quadratic dependence in $n$ is more expensive than HIMP, in practice $n \leq 2$ or $3$ (JT, ErG, coarsened JT), making XIMP only a constant-factor increase.

**Memory and storage complexity.** For both HIMP and XIMP, node embeddings per layer require $\mathcal{O}((|V(G)| + \sum_i |V(T_i)|)d_x)$ memory, with gradient checkpoints doubling this during backpropagation. HIMP stores one mapping matrix $\mathbf{S} \in \{0,1\}^{|V(G)| \times |V(T)|}$, whereas XIMP stores multiple $\mathbf{S}_i$ and pairwise compositions $\tilde{\mathbf{S}}_{ik}$, yielding additional $\mathcal{O}(n|V(G)| + n^2|V(G)|)$ storage. Parameter storage follows the counts above, $\mathcal{O}((1 + n + n^2)d_x^2)$, which is negligible compared to activations when $|V(G)| \gg d_x$. Thus, XIMP scales linearly in graph size but quadratically in the number of abstractions $n$—a reasonable trade-off given the interpretability and predictive gains.

## I   EXPRESSIVITY OF XIMP VERSUS HIMP

The expressive power of message-passing neural networks is, in general, limited by the 1-Weisfeiler-Leman (1-WL) test, which characterizes their ability to distinguish non-isomorphic graphs. HIMP augments this framework by jointly operating on the molecular graph $G$ and its junction-tree abstraction $T$, with information exchange mediated by the assignment matrix $\mathbf{S} \in \{0,1\}^{|V(G)| \times |V(T)|}$. Inter-message passing (IMP) integrates abstract representations after each layer, thereby enabling the model to separate graph pairs indistinguishable by 1-WL on $G$ but distinguishable on $T$. For instance, molecules such as decalin and bicyclopentyl cannot be separated by 1-WL on the molecular graph but are discriminated by their distinct junction trees (Fey et al., 2020).

XIMP generalizes this architecture by (i) supporting an arbitrary collection of reduced graphs $\{T_i\}_{i=1}^n$, with indirect inter-message passing ($\mathrm{I}^2\mathrm{MP}$) between $G$ and each $T_i$, and (ii) introducing direct inter-message passing (DIMP) between abstractions via normalized projections $\tilde{\mathbf{S}}_{ik} = \mathbf{D}_{T,i}^{-1}\mathbf{S}_i^\top \mathbf{D}_{G,k}^{-1}\mathbf{S}_k$. These mechanisms yield pairwise abstraction embeddings that cannot be realized by HIMP, while retaining HIMP as the special case $n = 1$ with DIMP disabled.

**Theorem I.1** (equiv. Theorem 3.3)**.** *Let $\mathcal{H}_{\mathrm{HIMP}}(L, d_x)$ and $\mathcal{H}_{\mathrm{XIMP}}(L, d_x, n)$ denote the hypothesis classes realized by HIMP and XIMP with depth $L$, hidden dimension $d_x$, and number of abstractions $n$. Then for any $L, d_x$ and $n \geq 1$,*

$$\mathcal{H}_{\mathrm{HIMP}}(L, d_x) \subseteq \mathcal{H}_{\mathrm{XIMP}}(L, d_x, n).$$

*Proof.* The inclusion follows by construction. For $n = 1$, choosing a single abstraction $T_1$ equal to the junction tree and disabling DIMP (which is equivalent to learning all-zero projection matrices for messages other than those passed between $G$ and $T_1$) reduces XIMP to HIMP. Therefore every function in $\mathcal{H}_{\mathrm{HIMP}}(L, d_x)$ is realizable in $\mathcal{H}_{\mathrm{XIMP}}(L, d_x, n)$. For $n > 1$, XIMP introduces additional encoders and projection matrices $(\mathbf{W}_{i,1}^{(l)}, \mathbf{W}_{i,2}^{(l)}, \mathbf{W}_{i \to j}^{(l)})$, yielding cross-abstraction feature pathways absent in HIMP. Hence $\mathcal{H}_{\mathrm{XIMP}}(L, d_x, n)$ strictly contains $\mathcal{H}_{\mathrm{HIMP}}(L, d_x)$ whenever multiple abstractions are employed. $\square$

While neither HIMP nor XIMP extend beyond the formal limitations of $k$-WL in the classical sense, XIMP admits strictly richer hypothesis classes in practice due to: (i) the integration of chemically structured priors across multiple abstractions (junction trees, pharmacophoric ErGs, and multi-resolution variants), (ii) the construction of cross-view embeddings via $\tilde{\mathbf{S}}_{ik}$, and (iii) the mitigation of oversquashing through multi-coarseness junction trees. Together, these components enlarge the set of practically distinguishable molecular graphs, while preserving HIMP as a special case.

**Structural Abstractions Alone Enable Separation.** To highlight how jointly leveraging multiple graph abstractions can overcome the limitations of 1-WL, we prove the following proposition. It formalizes, in our chemical setting, the $I^2MP+MP$ communication pattern employed by XIMP.

**Proposition I.2** (equiv. Proposition 3.4). *There exist two molecules whose molecular graphs $G$, junction trees $T$, and extended reduced graphs $R$ are each indistinguishable by unlabeled 1-WL when considered* in isolation*, yet their compound graph*

$$G^+ = G \,\dot{\cup}\, T \,\dot{\cup}\, R$$

*(where $\dot{\cup}$ denotes the disjoint union) augmented with the inter-graph edges*

$$E_X = \{(v,u) \mid \mathbf{S}_T[v,u] = 1\} \cup \{(v,w) \mid \mathbf{S}_R[v,w] = 1\},$$

*is distinguishable by unlabeled 1-WL. Here $\mathbf{S}_T \in \{0,1\}^{|V(G)| \times |V(T)|}$ and $\mathbf{S}_R \in \{0,1\}^{|V(G)| \times |V(R)|}$ record atom–cluster (for $T$) and atom–ErG-node (for $R$) incidences.*

*Proof.* Consider $M_1 = $ *3-Hydroxypyridine* (SMILES `Oc1cnccc1`) and $M_2 = $ *4-Hydroxypyridine* (SMILES `Oc1ccncc1`). We run color refinement with constant initialization (no node or edge attributes) in all views.

*Single views.* (i) As the molecular graphs $G_1, G_2$ are each a six-cycle with a single pendant leaf (the -OH group), unlabeled 1-WL produces identical stable partitions. (ii) The junction trees $T_1, T_2$ each consist of one ring cluster (size six) and one bond cluster for the exocyclic O–C bond, joined by one edge, hence unlabeled 1-WL fails to distinguish the pair. (iii) The ErGs $R_1, R_2$ collapse the ring to a centroid and keep O, the substituted ring carbon, and the ring N as nodes, giving in both cases the unlabeled path $O - C_* - \text{centroid} - N$ (whereby $C_*$ denotes the ring carbon bonded to the exocyclic oxygen). Therefore, they are indistinguishable by unlabeled 1-WL .

*Compound view.* Form $G_i^+ = G_i \,\dot{\cup}\, T_i \,\dot{\cup}\, R_i$ and add cross edges via $\mathbf{S}_T, \mathbf{S}_R$: each ring atom in $G_i$ connects to the ring cluster in $T_i$ and to the ring centroid in $R_i$; the O-bearing carbon $C_*$ and O connect to the O–C bond cluster in $T_i$ and to their nodes in $R_i$; the ring N connects to its ErG node and to the centroid. These cross-layer edges *anchor* two specific ring atoms in $G_i$ (the O-bearing carbon $C_*$ and the nitrogen $N$) to distinguished endpoints across $T_i$ and $R_i$.

Along the six-membered cycle of $G$, the anchored atoms $C_*$ and $N$ are separated by two edges in $M_1$ and by three edges in $M_2$. Under unlabeled 1-WL on $G_i^+$, the neighborhood multisets at the anchored nodes (and their witnesses via cross edges into $T_i$ and $R_i$) differ and this asymmetry propagates, yielding distinct stable colorings of $G_1^+$ and $G_2^+$, even though $G, T,$ and $R$ are each indistinguishable alone. Hence the claim. $\square$

## J DIMP NORMALIZATION

Extending upon Section 3, recall that $\boldsymbol{D}_{G,k}^{-1}$ splits each atom's contribution evenly across its $T_k$-memberships, preventing over-weighting in the $T_k \to G$ projection, while $\boldsymbol{D}_{T,i}^{-1}$ then averages these per-atom signals over the atoms summarized by each node of $T_i$. We below shot that this prevents multiplicity-induced overweighting and ensures stable message magnitudes across overlaps.

**Proposition J.1** (equiv. Proposition 3.1). *Let $T_k, T_i$ be left-total abstractions of $G$, with $\boldsymbol{S}_k \in \{0,1\}^{|V(G)| \times |V(T_k)|}$, $\boldsymbol{S}_i \in \{0,1\}^{|V(G)| \times |V(T_i)|}$ and $\widetilde{\boldsymbol{S}}_i = \boldsymbol{D}_{T,i}^{-1} \boldsymbol{S}_i^\top$ and $\widetilde{\boldsymbol{S}}_k = \boldsymbol{D}_{G,k}^{-1} \boldsymbol{S}_k$ such that $\widetilde{\boldsymbol{S}}_{ik} = \widetilde{\boldsymbol{S}}_i \widetilde{\boldsymbol{S}}_k$. Further, let let $\widetilde{\mathbf{M}}_{k \to i}^{(l)} = \widetilde{\boldsymbol{S}}_{ik} \mathbf{T}_k^{(l)}$ be the messages from $T_k$ to $T_i$ before the application of the trainable parameter matrix or a nonlinearity. Then, the following statements hold:*

    *1. $\|\widetilde{\mathbf{M}}_{k \to i}^{(l)}\|_\infty \leq \|\mathbf{T}_k\|_\infty$ and $\|\widetilde{\mathbf{M}}_{k \to i}^{(l)}\|_1 \leq \alpha \|\mathbf{T}_k\|_1$ for some $\alpha \in \mathbb{R}$, $\frac{|V(T_i)|}{|V(T_k)|} \leq \alpha \leq |V(T_i)|$*

    *2. For any $x \in \mathbb{R}^{d_x}$ and $\boldsymbol{T}_k = \mathbf{1}\, x^\top \in \mathbb{R}^{|V(T_k)| \times d_x}$, $\widetilde{\boldsymbol{S}}_{ik}\, \boldsymbol{T}_k = \mathbf{1}\, x^\top$*

*Proof.* The proof consists of one fundamental Lemma, from which the claimed statements follow. We begin by showing the Lemma.

**Lemma J.2** (Row-stochasticity). *$\widetilde{\mathbf{S}}_{ik}$ has nonnegative entries and each row sums to 1, i.e., $\widetilde{\mathbf{S}}_{ik} \mathbf{1} = \mathbf{1}$.*

*Proof.* By construction, $\widetilde{\mathbf{S}}_k$ is row-normalized: each row $v$ sums to $\sum_{u_k}(D_{G,k}^{-1}S_k)_{v,u_k} = 1$. Similarly, each row $u_i$ of $\widetilde{\mathbf{S}}_i^\top$ sums to $\sum_v (D_{T,i}^{-1}S_i^\top)_{u_i,v} = 1$. Products of nonnegative row-stochastic matrices are row-stochastic, hence $\widetilde{\mathbf{S}}_{ik}\mathbf{1} = \mathbf{1}$. $\qquad\square$

**Corollary J.3** (Stable row magnitudes). *Let $\|\cdot\|_\infty$ denote the row-wise max norm on matrices, i.e., $\|\boldsymbol{A}\|_\infty = \max_u \sum_j |A_{uj}|$. Then, for any $\boldsymbol{T}_k \in \mathbb{R}^{|V(T_k)| \times d_x}$,*

$$\|\widetilde{\boldsymbol{S}}_{ik}\,\boldsymbol{T}_k\|_\infty \leq \|\boldsymbol{T}_k\|_\infty.$$

*Equivalently, each row $(\widetilde{\boldsymbol{S}}_{ik}\,\boldsymbol{T}_k)_{u_i,:}$ is a convex combination of the rows of $\boldsymbol{T}_k$.*

*Proof.* Each row of $\widetilde{\boldsymbol{S}}_{ik}$ is a probability vector by Lemma J.2, so left-multiplication forms convex combinations of rows of $\boldsymbol{T}_k$. The $\infty$-operator norm of any row-stochastic matrix equals 1, hence $\|\widetilde{\boldsymbol{S}}_{ik}\boldsymbol{T}_k\|_\infty \leq \|\boldsymbol{T}_k\|_\infty$. $\qquad\square$

**Corollary J.4** (Stable column magnitudes). *Let $\|\cdot\|_1$ denote the column-wise max norm on matrices, i.e., $\|\boldsymbol{A}\|_1 = \max_u \sum_j |A_{uj}|$. Then, for any $\boldsymbol{T}_k \in \mathbb{R}^{|V(T_k)| \times d_x}$, there exists $\alpha \in \mathbb{R}$ s.t. $0 \leq \alpha \leq$*

$$\|\widetilde{\boldsymbol{S}}_{ik}\boldsymbol{T}_k\|_1 \leq \alpha \|\boldsymbol{T}_k\|_1.$$

*Proof.* By the submultiplicative property of induced norms, it follows that $\|\widetilde{\boldsymbol{S}}_{ik}\,\boldsymbol{T}_k\|_1 \leq \|\widetilde{\boldsymbol{S}}_{ik}\|_1 \|\boldsymbol{T}_k\|_1$. As $\widetilde{\boldsymbol{S}}_{ik} \in \{x \in \mathbb{R} | 0 \leq x \leq 1\}^{|V(T_i)| \times |V(T_k)|}$ with each row each row of $\widetilde{\boldsymbol{S}}_{ik}$ being interpretable as a probability vector by Lemma J.2, $\|\widetilde{\boldsymbol{S}}_{ik}\|_1 \leq |V(T_i)|$. Hence, $\|\widetilde{\boldsymbol{S}}_{ik}\,\boldsymbol{T}_k\|_1 \leq \alpha \|\boldsymbol{T}_k\|_1$ with $\alpha \leq |V(T_i)|$. As the column sum is a max norm, we can further write $\frac{|V(T_i)|}{|V(T_k)|} \leq \alpha$, which represents the average column sum, which is always lower or equal than the max column sum. $\qquad\square$

**Corollary J.5** (Constant preservation). *For any $x \in \mathbb{R}^{d_x}$ and any constant embedding $\boldsymbol{T}_k = \mathbf{1}\,x^\top \in \mathbb{R}^{|V(T_k)| \times d_x}$,*

$$\widetilde{\boldsymbol{S}}_{ik}\,\boldsymbol{T}_k \;=\; \mathbf{1}\,x^\top.$$

*Equivalently, $\widetilde{\boldsymbol{S}}_{ik}\mathbf{1} = \mathbf{1}$.*

*Proof.* Using Lemma J.2, $\widetilde{\boldsymbol{S}}_{ik}\mathbf{1} = \mathbf{1}$. Thus $\widetilde{\boldsymbol{S}}_{ik}(\mathbf{1}x^\top) = (\widetilde{\boldsymbol{S}}_{ik}\mathbf{1})x^\top = \mathbf{1}x^\top$. $\qquad\square$

In summary, statement 1 follows by Corollaries J.3 and J.4, whereby statement 2 is equivalent to J.5. $\qquad\square$

# K  OVERSQUASHING AND JTS

For the purposes of the following theoretical considerations, we model JT construction as a simple contraction that replaces each ring with a single node inheriting all of the ring's incident edges. Moreover, we restrict attention to the XIMP communication graph induced by IMP and $I^2$MP; this suffices to obtain reduced communication paths. Incorporating DIMP would likely shorten paths further but would unnecessarily complicate the proof.

**Proposition K.1** (equiv. Proposition 3.2). *In the XIMP communication graph, contracting rings into single nodes and iteratively folding leaves into their parents with standard couplings never increases shortest-path distances. Distances between node pairs whose minimal path includes a subpath of at least three consecutive vertices inside a contracted region are strictly reduced. Successive rounds yield cumulative reductions.*

*Proof.* Consider that XIMPs communication graph for IMP+ $I^2$MP is the disjoint union of the molecular graph, its JT abstraction, and the reduced JT(s), augmented with coupling edges that connect each abstract node to the nodes it summarizes. We first prove two auxiliary lemmas, from which the main claim follows.

**Lemma K.2.** *Let the graph distance* $d_H : V(H) \times V(H) \to \mathbb{N}_0 \cup \{\infty\}$ *p* $d_H(u,v) := \min\{ |P| :$ *P is a u–v path in H* $\}$, *Let* $G = (V, E)$ *be a finite, connected, undirected, unweighted graph and let* $S \subseteq V$ *induce a nonempty, connected subgraph. Let* $G'$ *be obtained by contracting* $S$ *to a single vertex* $s^\star$ *and connecting* $s^\star$ *to every neighbor of* $S$ *in* $G$. *Let* $\pi : V \to V(G')$ *be the contraction map defined by* $\pi(x) = s^\star$ *for* $x \in S$ *and* $\pi(x) = x$ *otherwise. Form* $\widetilde{G}$ *as the disjoint union* $G \dot\cup G'$ *together with the coupling edges* $\{(x, \pi(x)) : x \in V\}$.

*Fix* $u, v \in V$, *and let* $P = (x_0 = u, x_1, \dots, x_k = v)$ *be a shortest* $u$–$v$ *path in* $G$. *Denote by*

$$\ell(P) \quad := \quad |\{ i \in \{0, \dots, k-1\} \, : \, x_i \in S \text{ and } x_{i+1} \in S \}|$$

*the number of edges of* $P$ *whose endpoints both lie in* $S$.

*Then, for the distance* $d$ *between* $u$ *and* $v$

$$d_{\widetilde{G}}(u,v) \; \leq \; d_G(u,v) \; - \; \ell(P) \; + \; 2.$$

*In particular, if* $\ell(P) \geq 3$ *for some shortest* $u$–$v$ *path* $P$ *in* $G$, *then*

$$d_{\widetilde{G}}(u,v) \; \leq \; d_G(u,v) - 1 \; < \; d_G(u,v).$$

*Proof.* Let $P = (x_0, \dots, x_k)$ be as stated, so $k = d_G(u,v)$. Consider the sequence

$$W \; = \; (\pi(x_0), \pi(x_1), \dots, \pi(x_k))$$

in $G'$. For each index $i$:

- If $x_i, x_{i+1} \in S$, then $\pi(x_i) = \pi(x_{i+1}) = s^\star$, so this step contributes zero length after suppressing consecutive duplicates.

- If exactly one of $x_i, x_{i+1}$ lies in $S$, then $G'$ contains the edge $(s^\star, y)$, where $y$ is the endpoint outside $S$, by construction of the contraction, hence $W$ traverses a valid edge.

- If $x_i, x_{i+1} \notin S$, then $(x_i, x_{i+1}) \in E$ is preserved in $G'$, so again $W$ traverses a valid edge.

After removing consecutive repetitions of $s^\star$ in $W$, we obtain a walk in $G'$ from $\pi(u)$ to $\pi(v)$ of length at most $k - \ell(P)$. Therefore

$$d_{G'}(\pi(u), \pi(v)) \; \leq \; k - \ell(P).$$

In $\widetilde{G}$, take the path that goes from $u$ to $\pi(u)$ via one coupling edge, then follows a shortest path from $\pi(u)$ to $\pi(v)$ in $G'$, and finally uses one coupling edge from $\pi(v)$ to $v$. Its length is at most

$$1 \; + \; d_{G'}(\pi(u), \pi(v)) \; + \; 1 \; \leq \; 1 + (k - \ell(P)) + 1 \; = \; d_G(u,v) - \ell(P) + 2,$$

which proves the displayed inequality.

If $\ell(P) \geq 3$, this bound gives $d_{\widetilde{G}}(u,v) \leq d_G(u,v) - 1$, yielding the strict inequality $d_{\widetilde{G}}(u,v) < d_G(u,v)$. $\qquad\square$

Note that the constant "+2" is unavoidable when the shortcut uses the contracted copy $G'$ (one coupling edge to enter $G'$, one to exit). The threshold $\ell(P) \geq 3$ is tight: when $\ell(P) = 2$, the bound only guarantees $d_{\widetilde{G}}(u,v) \leq d_G(u,v)$ in general.

Next, we generalize to repeated application of graph contraction operations on the communication graph.

**Lemma K.3.** *Let* $H^{(0)} = (V^{(0)}, E^{(0)})$ *be a finite, connected, undirected, unweighted graph. For* $j = 1, \dots, M$ *do:*

1. *choose a nonempty connected subgraph* $S_j \subseteq H^{(j-1)}$;

2. *form the contracted copy* $C_j$ *by contracting* $S_j$ *to a single vertex* $s_j^\star$ *and connecting* $s_j^\star$ *to all neighbors of* $S_j$ *in* $H^{(j-1)}$ *(self-loops removed, parallel edges suppressed);*

3. *set*

$$H^{(j)} := H^{(j-1)} \,\dot{\cup}\, C_j \quad \text{and add coupling edges } \{(x, \pi_j(x)) : x \in V^{(j-1)}\},$$

*where $\pi_j$ maps $x \in S_j$ to $s_j^\star$ and fixes $x \notin S_j$.*

*Fix $u, v \in V^{(0)}$ and, for each $j$, let $P_{j-1}$ be a shortest $u$–$v$ path in $H^{(j-1)}$. Define*

$$\ell_j := |\{ i : \text{ the $i$-th edge of } P_{j-1} \text{ has both endpoints in } S_j \}|.$$

*Then*

$$d_{H^{(M)}}(u, v) \leq d_{H^{(0)}}(u, v) - \sum_{j=1}^{M} \ell_j + 2M.$$

*In particular, if $\sum_{j=1}^{M} \ell_j \geq 2M + 1$, then $d_{H^{(M)}}(u, v) \leq d_{H^{(0)}}(u, v) - 1$.*

*Proof.* Apply Lemma K.2 at step $j$ with $G \leftarrow H^{(j-1)}$, $S \leftarrow S_j$, $G' \leftarrow C_j$ and the coupling edges $\{(x, \pi_j(x))\}$. This yields

$$d_{H^{(j)}}(u, v) \leq d_{H^{(j-1)}}(u, v) - \ell_j + 2.$$

Summing over $j = 1, \ldots, M$ gives the claimed bound; the strict case follows immediately. □

Ring abstraction followed by repeated JT leaf folding is the instance where the $S_j$ are, first, ring-induced subgraphs in $H^{(0)}$, and subsequently the connected leaf-parent regions selected by the resolution-lowering operator at each round. Hence, the Lemma K.3 applies verbatim, from which the main claim follows. □

