# OpenReview forum: "XIMP: Cross Graph Inter-Message Passing for Molecular Property Prediction"
_ICLR.cc/2026/Conference — Submitted to ICLR 2026_

### Official Review · Reviewer_hMMy · 2025-10-17

**Soundness:** 3
**Presentation:** 3
**Contribution:** 2
**Rating:** 4
**Confidence:** 2

**Summary:**

The paper proposes XIMP, a generalization of HIMP that supports an arbitrary number of chemically interpretable graph abstractions (e.g., JT and ErG) alongside the molecular graph. XIMP introduces two cross-graph routes: indirect inter-message passing between the molecular graph and each abstraction, and direct inter-message passing among abstractions, combined with a learned multi-view readout. The model further employs multi-resolution JT coarsening to reduce information-flow distances and alleviate over-squashing. In experiments on ten diverse property-prediction tasks, XIMP typically matches or outperforms strong GNN baselines and fixed fingerprints.

**Strengths:**

S1: Empirically, XIMP is competitive or best on a majority of endpoints, with especially strong behavior on ADMET, and ablations isolate when DIMP/IMP and JT resolution matter.

S2: The presentation is very clear. The construction/mappings and message-passing flows are well illustrated—and the work is reproducible with code and detailed settings.

S3: The authors show that XIMP subsumes HIMP (a well-established method) both theoretically and experimentally.

**Weaknesses:**

W1: Complexity scaling with number of abstractions. Per-layer cost and parameters grow quadratically in hidden size and number of abstractions; experiments restrict to n smaller than 3.

W2: While 10 tasks are diverse, the evaluation is still centered on small/medium molecular benchmarks; protein-level or 3D-aware evaluations are left to future work.

W3: The authors tune the model using one kind of data split (stratified by labels), but they evaluate it using a different split (by chemical scaffolds). Because those two splits emphasize different generalization behaviors, the hyperparameters chosen during tuning may not work best on the actual test. The paper also reports “best-on-test” numbers (picking the settings that happen to score highest on the test set), which is an optimistic upper bound rather than a fair estimate.

W4: The claim that JT coarsening and cross-graph messaging alleviate over-squashing isn’t directly validated. The current benchmarks aren’t designed to diagnose long-range dependency failures, and there are no targeted stress tests (e.g., tunable bottlenecks or increasing shortest-path distances) to isolate the phenomenon.

**Questions:**

Q1: Could you report a validation protocol that better matches the scaffold test (e.g., group-k-fold by scaffold with label stratification within groups), and how XIMP fares relative to baselines under that scheme?

Q2: Do runtime/memory and accuracy scale smoothly when adding a third abstraction or deeper coarsening levels?

Q3: Could you empirically validate the claim that JT coarsening and cross-graph messaging alleviate over-squashing over known targeted benchmarks that isolate long-range dependencies?

---

### Official Review · Reviewer_KTZp · 2025-10-25

**Soundness:** 2
**Presentation:** 4
**Contribution:** 2
**Rating:** 2
**Confidence:** 4

**Summary:**

The paper proposes XIMP (cross graph inter-message passing), a message passing framework for molecular property prediction that extends MP-GNNs with additional graph abstractions and inter-graph message passing between them. Standard MP-GNNs pass messages on the molecular graph only. Prior work introduced HIMP (hierarchical inter-message passing) which augments this by introducing a single abstraction graph (e.g., Junction Tree or an extended reduced graph) and bidirectional message passing between the raw graph and that abstraction via learned couplings. XIMP takes the next step: it uses multiple abstractions in parallel and adds direct inter-abstraction message passing in addition to graph-to-abstraction passing. The paper claims improved robustness and accuracy across 10 tasks and offers structural arguments about expressivity and oversquashing.

**Strengths:**

- The paper is well written and presented. The method is described very clearly and the illustrations are an excellent help in understanding the process clearly.
- I like the overall idea of moving towards a large number of graph abstractions at once that all coordinate with each other. As a general approach I find this promising and there is large potential in combining this idea with orthogonal advances in GNN learning for molecules (that this submission unfortunately does not yet leverage).
- The relationship to the closely related prior work in HIMP is made very clear. Comparisons are made frequently and Theorem 3.3 clearly links expressivity of the two architectures.
- The reporting on experiments is very careful and conscious of how hyperparameter tuning is performed.

**Weaknesses:**

As a general point, the appraoch can be considered quite incremental over HIMP. While there is much potential to conceptually move far beyond HIMP, by pushing the idea of combining many abstractions further, I find that the submission does not sufficiently do so. It seems like a single incremental step was made, but I do not see evidence of this single step alone providing any significant benefits over HIMP, either theoretically or empirically.

## Oversquashing Analysis
I have major issues with the discussion around mitigating oversquashing. The analysis focuses on the intuitive correspondence to inter-node distance but  the depth of the analysis is lacking. The only stated result is Proposition 3.2, which is lacking in formality ("communication graph" and "folding leaves" are not defined and seemingly not used at all outside of that statement), but also it seems to be a trivial observation about contractions in graphs not increasing shortest paths. This has been a textbook observation for decades and I do not think it is appropriate to phrase this as a contribution of the submission.

Moreover, I think this analysis focuses on the wrong question. In my reading, it seems as if oversquashing might actually be introduced by the cross graph message passing. In particular, arbitrarily large cycles have all their nodes pass messages into a single node of the junction tree in the I$^2$MP part of XIMP (and in HIMP).

Ultimatley, with such a focus on oversquashing in the story of the paper I would have expected a deeper analysis of oversquashing behavior.

## Experimental Weaknesses
The reporting of the performacne of XIMP is misleading. The abstract says XIMP “outperforms state-of-the-art GNN and fingerprinting methods in most cases.” In the main tables, with validation-selected hyperparameters XIMP wins 4/10 tasks; HIMP also wins 4/10. I supose HIMP is conveniently considered neither a GNN or fingerprinting method. The advantage of XIMP still seems very minor with test-selected hyperparameters. On top of this, the tested GNNs are plain GCN, GAT, GraphSage and GIN, nowhere close to the state of the art on molecular benchmarks, e.g., CIN or any of the various ways to inject cycle information (local graph parameters/homomorphisms, loopy WL, etc.). Moreover, transformer approaches are missing entirely, despite their well documented strength on molecular tasks.

In summary, the experimental evaluation to me gives no clear indication of improvement. Performance is comparable to HIMP, which is in a way just a simpler version of XIMP and can be seen as an ablation of sorts. The comparison to GNNs is not informative as it does not compare to the state of the art. Comparison to transformer based approaches is missing.

**Questions:**

Why did you not compare to state of the art GNNs for molecular data or transformer based architectures?

---

### Official Review · Reviewer_tmWn · 2025-10-28

**Soundness:** 1
**Presentation:** 1
**Contribution:** 2
**Rating:** 2
**Confidence:** 4

**Summary:**

XIMP enhances performance by leveraging multi-level abstractions to mitigate oversquashing and improve long-range relational modeling. To achieve this, it enables flexible information exchange among abstractions while considering both direct and indirect message-passing pathways. This approach allows XIMP to learn effectively from chemically and structurally interpretable abstractions, demonstrating strong expressivity beyond the 1-WL test.

**Strengths:**

- **Oversquashing problem:** XIMP points out that previous studies have overlooked the oversquashing problem in molecular graphs and addresses it by introducing the DIMP and I²MP mechanisms.
- **performance across tasks:** XIMP generally outperforms the comparison models across various datasets and achieves the best results in the ECDF analysis.

**Weaknesses:**

- **Low readability:** The paper lacks clear logical flow, and the connection between the identified research gap and the proposed core idea is not presented naturally. In addition, the background description and related work are not clearly distinguished, making it difficult for readers to follow the overall structure. The definitions of key terms and concepts are also insufficient, which further reduces the overall clarity and readability of the paper.
- **Lack of comparison with models:** The study does not include comparisons with more recent or state-of-the-art models, limiting the strength of its empirical validation. As a result, it remains unclear whether XIMP offers advantages beyond existing graph-based architecture.
- **Limited methodological novelty:** The core idea of XIMP—inter-message passing across multiple graph abstractions—extends prior frameworks such as HIMP (Fey et al., 2020) and Finder et al. (2025) in a straightforward manner. The proposed architecture mainly generalizes existing message-passing schemes to multiple abstraction levels, without introducing a fundamentally new learning mechanism or theoretical insight.
- **Lack of experiments:** XIMP is proposed as a model to address the oversquashing problem; however, the experimental results do not provide sufficient evidence that it effectively alleviates this issue. Furthermore, in the MoleculeNet benchmark, which focuses on global-scale molecular properties, XIMP does not appear to show a significant performance improvement compared to conventional GNNs.

**Questions:**

- **Inconclusive empirical advantage:** In Table 1, XIMP achieves the best results on four datasets, which is the same as the GNN baseline, and the overall performance gap appears small. Could the authors clarify whether this difference is statistically significant? Also, the paper mentions a trade-off relationship, but such a trend is not clearly observable—could you elaborate on this point?
- **Clarification on Tables 1 and 2:** Tables 1 and 2 appear to present very similar results. Could the authors clarify what the main difference between these two tables is? In addition, the evaluation setting in Table 2—selecting the best hyperparameters on the test dataset—does not seem to be a common practice. Could the authors explain the reason for using this approach?
- **Question on oversquashing analysis:** The paper claims that XIMP mitigates oversquashing, but it is unclear whether this is empirically demonstrated. Are there experiments or analyses directly showing that XIMP reduces oversquashing, beyond overall performance improvements? It would also be helpful if the authors could explain how the proposed mechanism specifically contributes to alleviating oversquashing.
- **Clarification on Figure 5:** Could the authors clarify what Figure 5 represents and how it should be interpreted?

---

### Official Review · Reviewer_taPD · 2025-10-31

**Soundness:** 2
**Presentation:** 3
**Contribution:** 2
**Rating:** 2
**Confidence:** 4

**Summary:**

The work introduces XIMP (cross graph inter-message passing), which is a framework for passing messages both within and between graph abstractions. The authors claim that this new approach helps capture multi-level chemical structures and mitigate oversquashing. The paper also presents a series of experimental results on several different molecular property prediction datasets.

**Strengths:**

* In general, the paper is very well written and easy to follow.
* The idea of message passing both within and across abstractions is very interesting, and has the potential to generate more robust representations.

**Weaknesses:**

- My main concern is that the experimental results appear very weak. In particular:
    - XIMP is outperformed by other methods in more than half of the datasets evaluated. This weak performance indicates that there may not actually be any practical gain to the proposed method. Can the authors explain why this may be the case?
        - Furthermore, I find Table 2 to be misleading - picking the best hyperparameters based on a supposed “test set” invalidates its role as an unbiased evaluation set
        - The results in Figure 4 are also not very clear - is there a better/less cluttered way of presenting this?
    - The GNN baselines used for comparison are also not very strong. Since most of the analysis is focused on molecular property prediction, evaluating against stronger GNN/Graph Transformer models [1-5] that are known to perform well on molecules would strengthen the experimental claims.
    - The authors claim that this approach works beyond just the molecular domain, but they do not provide any evidence of that. Are there any experimental results to see how XIMP performs in different domains to support its generalisability?
- I’m not entirely convinced about the novelty of this work either, particularly with regards to the theoretical results.
    - Can the authors please comment on how their method meaningfully builds upon HIMP? And do they have any explanation as to why XIMP performs worse than HIMP if it’s supposed to be a generalisation of this previous method?
    - Proposition 3.2, which states that contracting nodes in a graph never increases shortest-path distance, is extremely trivial. Are there any other stronger results to support the author's claim that this approach mitigates oversquashing?

Overall, the paper does not seem to be fully fleshed out yet and requires more work. Therefore, I recommend it for rejection.

References:
1. Bodar et al. Weisfeiler and Lehman Go Cellular: CW Networks. In NeurIPS, 2021.
2. Rampášek et al. Recipe for a General, Powerful, Scalable Graph Transformer. In NeurIPS, 2022.
3. Jin et al. Homomorphism Counts for Graph Neural Networks. In ICML, 2024.
4. Luo et al. Enhancing Graph Transformers with Hierarchical Distance Structural Encoding. In NeurIPS, 2024.
5. Bao et al. Homomorphism Counts as Structural Encodings for Graph Learning. In ICLR, 2025.

**Questions:**

See weaknesses.

---

### Official Review · Reviewer_u3Kk · 2025-11-03

**Soundness:** 2
**Presentation:** 2
**Contribution:** 2
**Rating:** 4
**Confidence:** 4

**Summary:**

This manuscript proposes a new method called XIMP, a versatile inter-message-passing framework that learns over any number of arbitrary graph abstractions within a single model for molecular property prediction. The experimental results robustly demonstrate XIMP’s superior performance across ten diverse molecular property-prediction tasks; XIMP outperforms state-of-the-art GNNs and fingerprint baselines in most cases. The proposed approach is interesting and offers a new perspective for future research on molecular property prediction. However, the manuscript has several technical and detailed issues; more implementation details should be provided to ensure the replicability of the proposed approach.

**Strengths:**

1. The graph interaction algorithm is very innovative. Intermessage passing enables cross-graph exchange: information flows between the molecular graph and its abstractions and, in XIMP, also between different abstractions.
2. The inter-message passing incorporating the reduced graphs algorithm is very interesting, and the formula derivation part is very clear.

**Weaknesses:**

1. There are only two molecular graph abstractions in the main text. Please supplement with more flowcharts to enhance readability.
2. It is suggested to develop a free online web platform or local software to facilitate relevant biologists in conducting molecular property predictions.

**Questions:**

1. In Table 1, XIMP just yields the best predictive performance for the hyperparameters chosen via mean validation MAE in 4/10 cases. Please analyze the remaining 6 cases. Why is the model performing poorly in the remaining 6 cases?
2. In Tables 5 and 6, the compared GNNs are relatively old. Please add experiments to supplement the latest GNN model architecture.
3. In the ablation experiment section, the optimal selection of hyperparameters on the test set will naturally lead to data leakage. How to avoid this phenomenon.

---

### Meta-Review · Area_Chair_jSfP · 2026-01-01

**Summary:**

The work introduces XIMP (cross-graph inter-message passing), which is a framework for passing messages both within and between graph abstractions. The authors use XIMP to solve the molecular property prediction task.

Generally, the idea of cross-graph inter-message passing is not new, and it has been explored in existing studies, even for the molecular property prediction task (e.g., MV-GNN). This paper fails to discuss its novelty in relation to existing models. Additionally, the experimental baseline is outdated, which weakens the paper’s contributions.

Many other issues were raised by reviewers, and I believe these points cannot be addressed through rebuttals. Therefore, I recommend rejecting this paper.

**Reviewer Concerns:**

N.A.

**Reviewer Scores:**

N.A.

---

### Decision · Program_Chairs · 2026-01-26

Reject